# U2-BENCH: Benchmarking Large Vision-Language Models on Ultrasound Understanding

**Anjie Le**[*,1,2], **Henan Liu**[*,1,4], **Yue Wang**[10], **Zhenyu Liu**[1], **Rongkun Zhu**[5],
**Taohan Weng**[1,4], **Jinze Yu**[1,4], **Boyang Wang**[1,4], **Yalun Wu**[4], **Kaiwen Yan**[10],
**Quanlin Sun**[6], **Meirui Jiang**[1,7], **Jialun Pei**[7], **Siya Liu**[1], **Haoyun Zheng**[1], **Zhoujun Li**[4],
**J. Alison Noble**[2], **Jacques Souquet**[1,8], **Xiaoqing Guo**[†,2,5], **Manxi Lin**[†,9], **Hongcheng Guo**[†,1,3]

[1] Dolphin AI    [2] University of Oxford    [3] Fudan University    [4] Beihang University
[5] Hong Kong Baptist University    [6] University of Cambridge    [7] CUHK
[8] Esonic Imaging    [9] Technical University of Denmark    [10] Independent

## Abstract

Ultrasound is a widely-used imaging modality critical to global healthcare, yet its interpretation remains challenging due to variability in image quality caused by operator dependency, noise, and anatomical complexity. Although large vision-language models (LVLMs) have demonstrated impressive multimodal capabilities across natural and medical domains, their performance on ultrasound remains largely unexplored. We introduce U2-BENCH, the first comprehensive benchmark to evaluate LVLMs on ultrasound understanding across classification, detection, regression, and text generation tasks. U2-BENCH aggregates 7,241 cases spanning 15 anatomical regions and defines 8 clinically inspired tasks, such as *diagnosis*, *view recognition*, *lesion localization*, *clinical value estimation*, and *report generation*, across 50 ultrasound application scenarios. We evaluate 23 state-of-the-art LVLMs, both open- and closed-source, general-purpose and medical-specific. Our results reveal strong performance on image-level classification, but persistent challenges in spatial reasoning and clinical language generation. U2-BENCH establishes a rigorous and unified testbed to assess and accelerate LVLM research in the uniquely multimodal domain of medical ultrasound imaging. [1]

## 1 Introduction

Ultrasound (US) is one of the most widely used imaging modalities in global healthcare — essential in obstetrics, emergency medicine, cardiology, and low-resource settings — while its interpretation remains notoriously difficult (Hewson & Bedforth, 2023). Compared to modalities such as computed tomography (CT), magnetic resonance imaging (MRI), positron emission tomography (PET), and whole-slide imaging (WSI), which offer higher spatial resolution, consistent image quality, and standardized anatomical views, ultrasound is real-time and low-cost but highly operator-dependent and frequently affected by imaging artifacts (Sharma et al., 2021). In addition, in contrast to these modalities, US is dynamically presenting three-dimensional (3D) anatomies in image sequences. Therefore, accurate interpretation of US demands not only visual pattern recognition in the images, but also an understanding of anatomy and capturing of dynamic spatial-context reasoning, typically requiring extensive prior domain expertise (Wang et al., 2022). These challenges have limited the applicability of earlier artificial intelligence (AI) models. However, recent advances in medical large vision-language models (LVLMs) have shown promise in overcoming these barriers (Chen et al., 2024b; Xia et al., 2024; Huang et al., 2025), potentially offering a robust multimodal understanding of complex, noisy, and context-rich ultrasound data.

---

[1] Datasets and code are available at: `https://huggingface.co/datasets/DolphinAI/u2-bench/tree/main`. The leaderboard is available at: `https://dolphin-sound.github.io/u2-bench/`.

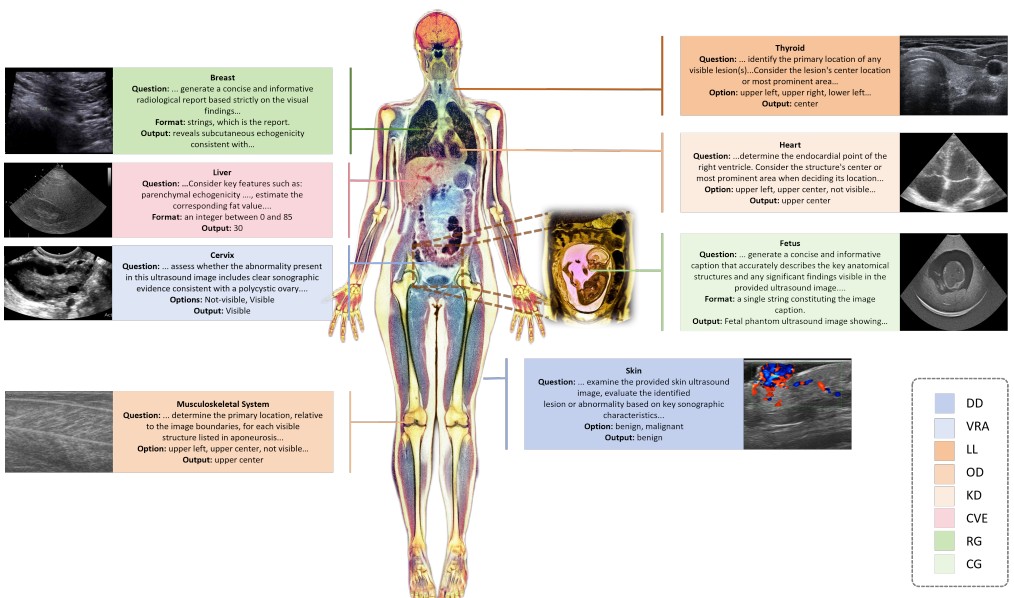

Figure 1: **Examples of the 8 benchmark tasks in U2-BENCH across diverse anatomical regions.** Each callout, consisting of the question prompt, expected output format, and sample output, highlights a representative ultrasound application scenario of the corresponding task. Tasks involve disease diagnosis (DD), view recognition and assessment (VRA), lesion localization (LL), organ detection (OD), keypoint detection (KD), clinical value estimation (CVE), report generation (RG) and caption generation (CG).

While progress in medical LVLM has been rapid, most previous models and benchmarks focus on those less noisy and static imaging modalities (Ji et al., 2022; Huang et al., 2023; Sivasubramaniam et al., 2024), leaving the complexities of ultrasound largely unaddressed. Prior efforts in ultrasound AI are typically based on small, task-specific datasets (Xiao et al., 2025), such as fetal plane identification (Guo et al., 2024) or pathology segmentation (Indelman et al., 2024; Ravishankar et al., 2023). As model capabilities grow, a public, balanced benchmark for ultrasound understanding is needed to evaluate whether emerging LVLMs can generalize beyond static medical vision tasks, to those requiring spatial reasoning and contextual understanding of anatomical structures.

To address these challenges, we introduce **U2-BENCH**, the first benchmark holistically evaluating current LVLMs for ultrasound understanding across diverse tasks and anatomies. The dataset we use comprises 7,241 cases across 15 anatomical regions, involving breast, heart, lung, etc, covering 8 diverse clinical tasks and 50 application scenarios. Each task belongs to one of the four categories: (1) classification (i.e., disease diagnosis, view recognition and assessment), (2) detection (i.e., lesion localization, organ detection, keypoint detection), (3) regression (i.e., clinical value estimation), (4) text generation (i.e., report generation, caption generation). Samples are selected to ensure balance across data sources, anatomies, and task types, to enable robust evaluation and alleviate dataset-specific bias. Several examples in our **U2-BENCH** are shown in Fig. 1.

We benchmark 20 LVLMs, including both open- and closed-source, general-purpose and medical-specialized models, on a diverse set of US tasks. **U2-BENCH** makes the following key contributions:

- **Comprehensive Dataset:** We release the first publicly available benchmark comprising 7,241 ultrasound cases spanning 15 anatomies and 8 clinical tasks, covering 50 application scenarios. Each case is annotated with task-aligned labels in a unified format and paired with carefully designed prompts, enabling standardized and reproducible evaluation.

- **Task Suite and Evaluation:** We define an eight-task taxonomy spanning *disease diagnosis*, *view recognition and assessment*, *lesion localization*, *organ detection*, *keypoint detection*, *clinical value estimation*, *report generation*, and *caption generation*. Each task reflects real-world clinical workflows and is paired with standard evaluation metrics. We also introduce an aggregate metric to provide a unified assessment of each model's overall capability in ultrasound understanding.

- **Empirical Insights:** We conduct the first large-scale evaluation of LVLMs on ultrasound, uncovering consistent trends across model families: models achieve strong performance on image-level disease diagnosis and clinical value estimation tasks, but degrade on spatial reasoning tasks such as view recognition and organ detection. Clinical report generation tasks remain particularly challenging. Performance gains from model scaling can be limited, and compact models occasionally outperform larger ones on certain tasks, suggesting that targeted training may be more impactful than scale alone in ultrasound understanding.

## 2 RELATED WORK

**Large Vision-Language Models.** LVLMs such as GPT-4V (OpenAI, 2023), Claude (Anthropic, 2024), Gemini (Anil et al., 2023), DeepSeek-VL (DeepSeek-AI et al., 2024), LLaVA (Liu et al., 2023a), Qwen-VL (Bai et al., 2023b), and MiniGPT4 (Zhu et al., 2023) have emerged as general-purpose multimodal systems capable of handling tasks like image captioning, visual question answering, and multimodal reasoning. These models are trained on large-scale image-text pairs (Sharma et al., 2018; Schuhmann et al., 2022), and their performance has been extensively evaluated in domains such as question answering, mathematics, and science (Chen et al., 2021; Sun et al., 2023; Wang et al., 2023; Huang et al., 2022; Liu et al., 2023b). However, their clinical reliability remains underexplored.

To address this gap, several medical-specialized LVLMs have been proposed. MiniGPT-Med (Wu et al., 2023b) focuses on X-ray, CT, and MRI for tasks such as medical report generation, VQA, and disease identification. RadFM (Wu et al., 2023a) further supports both 2D and 3D modalities. MedDr (He et al., 2024) extends to radiology, pathology, dermatology, retinography, and endoscopy, introducing a retrieval-augmented diagnosis strategy. Lingshu (Xu et al., 2025) is a recent medical LVLM that covers multiple imaging modalities. Yet, these models exclude ultrasound. Med-Gemini (Team, 2024) and MedGemma (Sellergren et al., 2025) span numerous modalities including ultrasound, though their capability in this domain is limited to caption generation.

**Multimodal Benchmarks for Large Vision-Language Models.** Several benchmarks assess general-domain LVLMs. MMBench (Liu et al., 2023c), MMT-Bench (Ying et al., 2024), and SEED-Bench (Li et al., 2023a) evaluate general-domain LVLMs through bilingual multiple-choice questions, large-scale visual reasoning tasks, and generative comprehension across image and video VQA, respectively. However, these benchmarks emphasize general-purpose visual understanding and omit clinically grounded evaluation.

Early medical VQA datasets like VQA-RAD (Lau et al., 2018), VQA-Med (Ben Abacha et al., 2019), and PathVQA (He et al., 2020) offer radiology or pathology image–question pairs but are not designed for evaluating LVLMs. GMAI-MMBench (Chen et al., 2024a) introduces a large-scale VQA-style benchmark for medical LVLMs, yet it contains only about 1.4k ultrasound cases primarily focused on classification and segmentation on 6 anatomies, and does not evaluate broader model capabilities such as clinical value estimation or structured report generation. In contrast, our **U2-BENCH** focuses exclusively on ultrasound and includes a diverse set of clinically meaningful tasks and anatomical regions. We have also included a comparison with existing ultrasound foundational datasets in Appendix A.

## 3 U2-BENCH

**Overview.** **U2-BENCH** is designed to holistically assess the capabilities of LVLM in ultrasound tasks. Section 3.1 introduces the eight clinically inspired tasks involved in our evaluation, which reflect essential diagnostic and reasoning abilities in ultrasound understanding. Section 3.2 details our benchmark construction pipeline, including dataset curation, preprocessing, and task-specific prompting. Section 3.3 summarizes the statistical property of the resulting dataset, which comprises 7,241 cases across 15 anatomies.

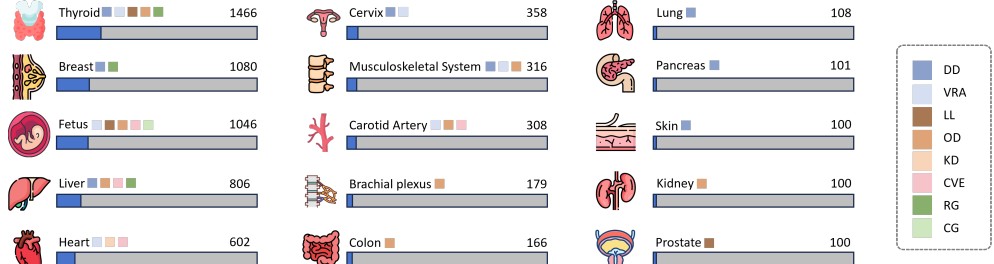

Figure 2: **Distribution of benchmark tasks across 15 anatomical regions in U2-BENCH.** The colored boxes next to each anatomy name indicate the benchmark tasks available for that anatomy, with each color corresponding to one of the eight core tasks (legend shown on the right). The blue bar represents the total number of samples for each anatomy region, with its length proportional to the sample count. Multiple tasks may share samples from the same anatomical region, depending on annotation availability and clinical relevance.

## 3.1 TASK DEFINITIONS

**U2-BENCH** focuses on four core capabilities: classification, detection, regression, and text generation, to systematically evaluate the performance of LVLMs on ultrasound-related tasks. We define eight tasks based on common ultrasound use cases, designed to probe a range of multimodal abilities, including anatomy recognition and clinical reporting. The task set was informed by typical sonography workflows and refined with input from domain experts to ensure practical relevance. Together, these tasks provide a structured benchmark for assessing LVLM performance across diverse ultrasound application scenarios. The eight tasks are as follows:

**Disease Diagnosis (DD).** This task requires the model to identify the presence and severity of a disease condition, such as grading in the Breast Imaging Reporting and Data System, based on the appearance of the ultrasound image. The task evaluates the ability of LVLMs to extract high-level semantic features and generate clinically aligned diagnostic predictions.

**View Recognition and Assessment (VRA).** In clinical practice, accurate diagnosis relies on the clear presentation of anatomical structures from specific angles, referred to as ultrasound standard planes. This task evaluates the ability of a model to assess image quality and classify scans into standard planes corresponding to different anatomical structures, such as the fetal head or abdominal long axis.

**Lesion Localization (LL).** Given a diagnostic image, the LVLM is asked to identify the location of a lesion, such as a suspicious breast mass, by selecting from nine predefined spatial categories such as upper left, center, or lower right. This task evaluates the spatial reasoning, saliency alignment, and ability to detect subtle structural abnormalities of LVLMs.

**Organ Detection (OD).** This task involves identifying the presence and boundaries of target organs in the ultrasound field of view, such as liver, kidney, or nerve. It assesses coarse-grained visual recognition under challenges unique to ultrasound, such as acoustic shadowing, inter-patient variability, and orientation ambiguity from manual probe handling.

**Keypoint Detection (KD).** In measurement tasks such as fetal biometry and adult echocardiography, precise localization of anatomical landmarks is critical for deriving clinically meaningful measurements. This task evaluates the fine-grained spatial understanding and geometric reasoning ability of the model, which are essential for tasks like skeletal length and chamber size estimation.

**Clinical Value Estimation (CVE).** This task involves predicting continuous clinical parameters derived from ultrasound images, such as lesion size, left ventricular ejection fraction, or liver fat percentage. It covers both anatomical and functional indicators relevant to diagnosis, treatment planning, and longitudinal monitoring, and evaluates whether the model can perform image-to-value regression by mapping visual inputs to clinically meaningful quantitative outputs.

**Report Generation (RG).** The model is prompted to generate a structured clinical report based on visual input, following the format of example reports provided in the prompt. This task evaluates the ability of LVLM to perform medical language generation and produce outputs that align with standard ultrasound reporting practices.

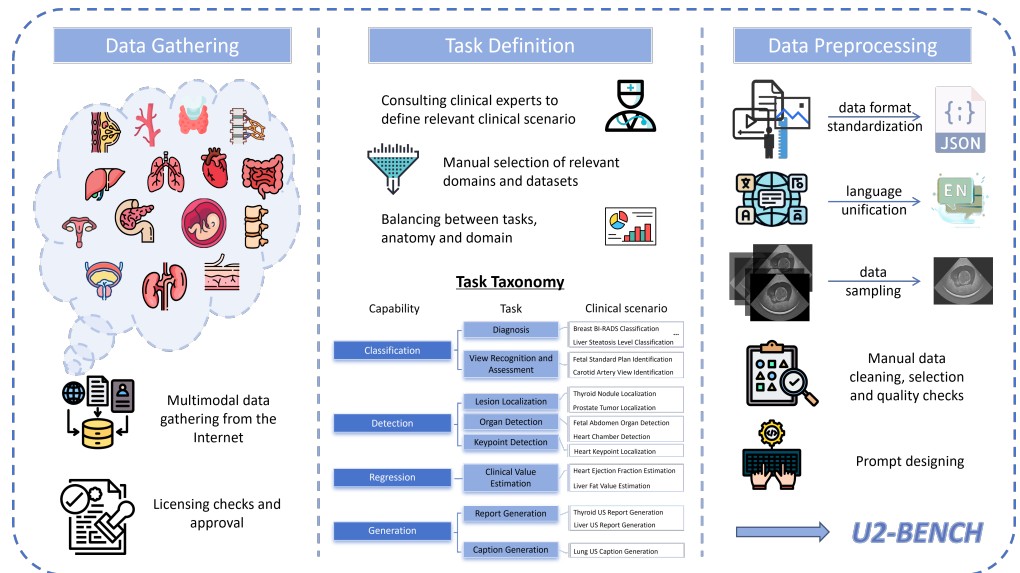

Figure 3: **Overview of the U2-BENCH construction pipeline.** The benchmark is built through three stages: (1) data gathering from 40 licensed ultrasound datasets spanning 15 anatomical regions, (2) task definition across 8 clinically inspired tasks grouped into four core capabilities: classification, detection, regression, and text generation, (3) data preprocessing, including annotation standardization, metadata unification, image/frame selection, and quality verification. This unified pipeline ensures benchmark consistency and clinical relevance across diverse ultrasound scenarios.

**Caption Generation (CG).** The model is asked to generate a concise anatomical description of a diagnostic image, guided by example captions provided in the prompt. This task evaluates basic visual-language alignment and the ability to verbalize structural features in a clinically appropriate manner of LVLM.

## 3.2 DATA CURATION AND PROCESSING

In this section, following the approach of previous benchmark constructions (Chen et al., 2024c; Xu et al., 2023; Zhong et al., 2023), we outline the three key steps used to build **U2-BENCH**: (1) data collection and sampling (2) data cleaning, format unification and quality verification, and (3) task-specific prompt design. Figure 3 summarizes the data processing pipeline.

**Data Selection and Sampling.** We construct **U2-BENCH** by sampling 7,241 ultrasound studies from 40 licensed datasets. In particular, 25 out of the 40 datasets were initially identified and compiled by Alsharid et al. (2025). For a more comprehensive dataset-level characterization of that data, we refer readers to that paper.

These datasets were selected to represent a wide range of diagnostic tasks, anatomical regions, and clinical contexts. While the original datasets were independently curated and clinically annotated, we performed standardization, sampling, and quality checks to ensure consistency across tasks and enable reliable, reproducible benchmarking. Some datasets contribute to multiple benchmark tasks based on their available annotations and clinical relevance.

To reflect real-world clinical data distributions and prevent data leakage, we adopt a task-specific, patient-level sampling strategy. Sampling is performed at the subject level rather than the image level to preserve intra-patient consistency. Importantly, during the sampling stage, datasets corresponding to clinically high-priority but data-sparse tasks were intentionally oversampled based on guidance from collaborating clinicians. To ensure anatomical coverage, we include data from 15 anatomical regions: fetus, thyroid, breast, heart, liver, cervix, carotid artery, musculoskeletal system, kidney, prostate, skin, lung, pancreas, brachial plexus, and colon.

**Data Cleaning, Format Unification, and Quality Verification.** All data in **U2-BENCH** are standardized into a unified format to support consistent parsing and evaluation across the dataset.

Ultrasound scans are converted to a uniform image format. For video sequences, a small number of representative frames are sampled per study to control evaluation cost while retaining key diagnostic content. Task-relevant metadata, including anatomy labels, measurements, and reports, is preserved in a structured schema. Segmentation masks are converted to bounding boxes. Texts are translated into English using a medically guided translation pipeline, with ambiguous terms resolved via a curated glossary and final verification by clinicians.

To ensure the reliability of **U2-BENCH**, we adopt both automated and manual quality assurance procedures during data preparation.

**(1) Automated Filtering.** During data preprocessing, we systematically check for missing labels, inconsistent or invalid annotations, and corrupted or unreadable files. Samples that fail these checks are discarded.

**(2) Manual Verification.** A team of 10 annotators manually reviewed all cases using a cross-validation protocol, where each data point was independently assessed by at least three annotators. Specifically, an engineer first check the validity of the metadata in the json file, then a biomedical expert independently checked for label–image consistency, measurement units and standardized anatomical terminology. A clinician then performed a final review of the diagnostic consistency on all processed cases while writing the task-specific prompts.

**Task-Specific Prompt Designing.** To ensure consistent model behavior and fair comparability across tasks, we design structured prompts for each of the 50 application scenarios, consisting of three components: (1) a clinical role definition to set context and expertise, (2) a task-specific instruction aligned with standard sonography workflow, and (3) an output format specification, such as classification options, value ranges, or reference output examples. Detailed prompts are included in Appendix D. An ablation study on the impact of prompt design is presented in Section 5.2.

## 3.3 STATISTICS

**U2-BENCH** comprises 7,241 ultrasound studies spanning 8 benchmark tasks and 15 anatomical regions. Table 5 in Appendix C details the number of cases per task. Classification and detection constitute the largest shares, with 2,999 and 2,921 cases, respectively, while generation and regression tasks provide targeted evaluation of report synthesis and clinical value estimation.

Figure 2 summarizes the distribution across anatomical regions. Thyroid and breast ultrasound together account for more than one-third of all cases. This is because of their high clinical prevalence and broad diagnostic utility. Many anatomies support multiple tasks - for instance, fetal ultrasound is used for classification and regression - enabling multi-task evaluation within a unified anatomical context. This composition ensures broad coverage across modalities, tasks, and body regions, supporting robust and clinically grounded assessment of LVLM performance.

## 4 EXPERIMENT

### 4.1 EVALUATION SETTINGS

We evaluated **U2-BENCH** on 20 LVLMs, both open-source and closed-source. A detailed list of the exact model versions evaluated and additional experimental details are provided in Appendix C, with the full implementation and hyperparameter configurations available in our public code repository. Detailed prompts are given in Appendix D.

### 4.2 EVALUATION PROTOCOL

We employed standard metrics aligned with clinical relevance and prior LVLM benchmarks. Classification tasks were evaluated with accuracy and F1 score. For detection-related tasks, we initially evaluated models using the ground-truth bounding box or coordinate outputs. However, many LVLMs failed to reliably generate valid coordinates or follow bounding-box formatting instructions. To enable stable and comparable evaluation across models, we therefore simplified the detection tasks into a 9-class position-classification formulation, where each region corresponds to a coarse spatial sector of the image. Under this formulation, detection tasks utilized accuracy as the metric to

assess localization correctness. Regression tasks report Root Mean Squared Error (RMSE), Mean Absolute Error (MAE), and percentage within tolerance (%_tol). Generation tasks were assessed with BLEU-4 as percentage, ROUGE, and BERTScore (Zhang* et al., 2020) to capture both lexical and semantic similarity. All metrics were computed using ground-truth labels from the original dataset and standardized outputs with the format specified by the prompts across models to ensure fair comparison.

**U2-Score.** We design a quantitative score to provide an overall evaluation metric for the ultrasound understanding capability of a model. The **U2-SCORE** is defined as a weighted combination of the metrics across all tasks, which is mathematically equivalent to computing a case-level average, consistent with prior work (Chen et al., 2024a). This can be formulated as:

$$\text{U2-Score} := \sum_{t=1}^{N} w_t d_t, \text{ where } w_t = \frac{n_t}{\sum_j n_j}, \text{ and } d_t \leq 1 \tag{1}$$

where $N$ represents the number of tasks, $w_t$ is the corresponding task weight, which is computed from the proportion of the sample number $n_t$ of the $t$-th task. This can mitigate the imbalance issue of sample size in different tasks. Here, $d_t$ denotes the value of the selected metric of the $t$-th task. More details are included in Appendix C.

## 4.3 EVALUATION RESULTS

We present a comprehensive comparison of multimodal models on the **U2-BENCH** benchmark (Table 1), aiming to identify key performance trends across tasks and model types. A more detailed example cases and error analysis is included in Appendix C.

**Closed-Source Models Lead.** Closed-source models continue to dominate, with **Dolphin-V1** achieving the highest overall score of **0.5835**, substantially outperforming all other models. The next strongest proprietary model, **GPT-5**, reaches a U2-Score of **0.3250**, followed by **Gemini-2.5-Pro-Preview** at **0.2968**. While the best open-source model, **DeepSeek-VL2**, attains a competitive score of **0.2630**, the gap to closed-source systems remains significant. These results highlight that despite rapid advances in open-source approaches, closed-source models still benefit from access to larger proprietary datasets and tailored optimization, giving them a clear performance edge.

**Task Difficulty Varies Significantly.** Image classification tasks remain the most tractable, with **Dolphin-V1** achieving the highest accuracy of **0.682** on **DD**, and several other models exceeding 0.48. In contrast, spatial reasoning and text generation remain difficult: no model surpasses **0.160** accuracy on **KD**, and all models fall below **7.5 BLEU** on **RG**. Regression tasks are also challenging; only the closed-source **Qwen-Max** reduces RMSE to **0.1248**, while all open-source models remain above **0.1675**.

**Scaling Brings Diminishing Returns.** Within the **Qwen-2.5-VL** family, scaling from 3B to 72B parameters yields consistent performance gains. While larger models achieve lower **CVE** RMSE, improvements in language generation and spatial reasoning tasks plateau, suggesting that excessive scaling may lead to overfitting on superficial visual patterns, ultimately harming clinical text generation capabilities.

**Domain-Specific Models Excel in Reasoning.** Medical-domain models such as **MedDr** show competitive performance on reasoning tasks (e.g., **CVE** RMSE = 0.214; **CG** BERT = 81.21), outperforming many general-purpose systems in structured clinical evaluation. Similarly, **MedGemma-4B-it** achieves the second-best CVE performance (RMSE = 0.167), highlighting the advantage of domain adaptation for quantitative reasoning. However, these models still lag behind larger general multimodal models on visual classification. For example, **Qwen-72B** achieves a DD F1 of 0.456, compared to MedDr's 0.312. This suggests that domain-specialized models are particularly effective for semantic and reasoning-heavy tasks, while general-purpose models retain an edge in coarse-grained visual recognition.

Table 1: Results of different models on the **U2-BENCH**. We utilize green (1st), blue (2nd), and yellow (3rd) backgrounds to distinguish the top three results within different models. The "U2-Score" column represents the quantitative score defined in Section 4.2. To calculate the **U2-SCORE** for random guessing, the BLEU scores are taken to be zero.

| Models | DD Acc. ↑ | DD F1 ↑ | VRA Acc. ↑ | VRA F1 ↑ | LL Acc. ↑ | OD Acc. ↑ | KD Acc. ↑ | CVE RMSE ↓ | CVE MAE ↓ | CVE %_tol ↓ | RG BLEU% ↑ | RG Rouge% ↑ | RG BERT% ↑ | CG BLEU% ↑ | CG Rouge% ↑ | CG BERT% ↑ | U2-Score ↑ |
|---|---|---|---|---|---|---|---|---|---|---|---|---|---|---|---|---|---|
| Random Guessing | 0.4143 | 0.4135 | 0.3195 | 0.3184 | 0.1118 | 0.0680 | 0.1120 | 0.5472 | 0.4352 | 18.776 | - | - | - | - | - | - | - |
| *Medical-Specific Models* | | | | | | | | | | | | | | | | | |
| MiniGPT-Med | 0.3468 | 0.2828 | 0.1800 | 0.1048 | 0.1728 | 0.1789 | 0.0840 | 0.3056 | 0.2600 | 33.2259 | 6.4700 | 20.1300 | 74.6900 | 30.2000 | 47.7500 | 80.5000 | 0.2375 |
| MedDr | 0.4508 | 0.3118 | 0.2071 | 0.1214 | 0.0720 | 0.0881 | 0.0900 | 0.2144 | 0.1786 | 38.2642 | 2.7998 | 13.5060 | 72.2050 | 33.4939 | 49.6236 | 81.2078 | 0.2373 |
| MedGemma-4B-it | 0.5005 | 0.4336 | 0.3071 | 0.1520 | 0.2750 | 0.0858 | 0.0200 | 0.1667 | 0.1316 | 55.0962 | 1.5360 | 15.0348 | 74.0205 | 4.8777 | 35.9803 | 76.7859 | 0.2668 |
| Lingshu-7B | 0.4589 | 0.2755 | 0.2625 | 0.1490 | 0.1265 | 0.2005 | 0.1140 | 0.2581 | 0.1908 | 27.8302 | 1.9974 | 15.7764 | 67.8138 | 4.0058 | 12.3106 | 62.0800 | 0.2704 |
| *Open-Source Multimodal Models* | | | | | | | | | | | | | | | | | |
| Qwen-2.5-VL-3B-Instruct | 0.4503 | 0.3591 | 0.2097 | 0.1492 | 0.0696 | 0.0649 | 0.0894 | 0.5008 | 0.4519 | 18.9055 | 3.5018 | 15.0327 | 72.8419 | 27.6748 | 44.7618 | 79.8849 | 0.2095 |
| Qwen-2.5-VL-7B-Instruct | 0.4821 | 0.3860 | 0.2181 | 0.1665 | 0.0750 | 0.0704 | 0.1000 | 0.4646 | 0.4337 | 19.7115 | 3.7100 | 15.5600 | 73.1500 | 29.4400 | 47.0000 | 81.1500 | 0.2235 |
| Qwen-2.5-VL-32B-Instruct | 0.4812 | 0.3860 | 0.2864 | 0.2071 | 0.1700 | 0.0755 | 0.0880 | 0.3414 | 0.3015 | 27.4038 | 1.1900 | 13.0100 | 68.1400 | 14.7700 | 38.6800 | 77.3900 | 0.2449 |
| Qwen-2.5-VL-72B-Instruct | 0.4895 | 0.4556 | 0.2559 | 0.1789 | 0.1150 | 0.0660 | 0.0860 | 0.3224 | 0.2733 | 37.9370 | 3.0900 | 15.0600 | 72.6600 | 28.1600 | 44.2800 | 80.9100 | 0.2421 |
| DeepSeek-VL2 | 0.4126 | 0.3190 | 0.2268 | 0.1111 | 0.2950 | 0.1682 | 0.1320 | 0.2956 | 0.2505 | 12.3355 | 7.4700 | 20.5400 | 75.3800 | 11.4200 | 34.8500 | 77.2400 | 0.2630 |
| InternVL3-9B-Instruct | 0.4447 | 0.3716 | 0.1926 | 0.1083 | 0.3000 | 0.1416 | 0.0940 | 0.2429 | 0.1733 | 50.8738 | 2.1600 | 14.7000 | 72.2100 | 21.5900 | 43.1300 | 80.9800 | 0.2566 |
| LLaVA-1.5-13B | 0.4321 | 0.3055 | 0.1731 | 0.0755 | 0.1700 | 0.1259 | 0.1100 | 0.2307 | 0.1976 | 24.7964 | 6.2400 | 18.5800 | 73.7900 | 10.8300 | 29.4000 | 75.5000 | 0.2378 |
| Phi-4-Multimodal-Instruct | 0.3686 | 0.1148 | 0.2452 | 0.0537 | 0.0350 | 0.0815 | 0.1600 | 0.2249 | 0.2006 | 16.1972 | 3.2700 | 16.5800 | 73.2700 | 3.8700 | 22.9800 | 73.0800 | 0.2168 |
| Mistral-Small-3.1-24B-Inst. | 0.4359 | 0.0936 | 0.1964 | 0.0664 | 0.1300 | 0.0910 | 0.1060 | 0.1675 | 0.1331 | 45.9459 | 1.8000 | 14.9000 | 71.7200 | 20.7700 | 42.1200 | 80.7400 | 0.2356 |
| *Closed-Source Multimodal Models* | | | | | | | | | | | | | | | | | |
| Doubao-1.5-Vision-Pro-32k | 0.5580 | 0.2597 | 0.2922 | 0.2147 | 0.1700 | 0.0729 | 0.1240 | 0.3664 | 0.3377 | 33.1731 | 0.7100 | 6.6450 | 72.4000 | 8.6400 | 33.3000 | 78.4200 | 0.2587 |
| GPT-4o-Mini | 0.4924 | 0.3784 | 0.1922 | 0.1272 | 0.1357 | 0.0846 | 0.0960 | 0.2267 | 0.1976 | 19.2308 | 4.9400 | 17.5200 | 74.1300 | 11.7300 | 36.2900 | 77.5300 | 0.2388 |
| GPT-4o | 0.4928 | 0.4132 | 0.1504 | 0.0974 | 0.1161 | 0.0850 | 0.0960 | 0.3712 | 0.3527 | 15.7895 | 2.6800 | 14.7700 | 73.3500 | 33.7700 | 49.9600 | 81.5800 | 0.2253 |
| GPT-5 | 0.5366 | 0.4590 | 0.4573 | 0.3550 | 0.2662 | 0.1767 | 0.1080 | 0.3097 | 0.1878 | 36.1867 | 1.0641 | 8.7440 | 66.8302 | 7.9669 | 23.3116 | 72.2203 | 0.3250 |
| Gemini-1.5-Pro | 0.3781 | 0.2247 | 0.0909 | 0.0476 | 0.2700 | 0.0661 | 0.0980 | 0.2772 | 0.2205 | 40.7051 | 0.5800 | 9.9400 | 70.5500 | 28.5800 | 45.9200 | 80.0200 | 0.1999 |
| Gemini-2.0-Pro-Exp | 0.4925 | 0.4194 | 0.1648 | 0.1323 | 0.1714 | 0.0820 | 0.1945 | 0.1945 | 0.1498 | 53.3333 | 0.2600 | 6.9200 | 40.2400 | 31.1800 | 48.6000 | 81.6000 | 0.2438 |
| Gemini-2.5-Pro-Preview | 0.4256 | 0.3112 | 0.2098 | 0.1493 | 0.2709 | 0.2714 | 0.2518 | 0.2937 | 0.2672 | 34.4970 | 5.5030 | 18.0180 | 74.4930 | 15.0110 | 38.0070 | 75.9890 | 0.2968 |
| Claude-3.7-Sonnet | 0.2121 | 0.0449 | 0.1453 | 0.0479 | 0.1356 | 0.0540 | 0.0760 | 0.1764 | 0.1500 | 36.0215 | 0.6900 | 12.2300 | 68.7400 | 1.2900 | 16.6600 | 71.6600 | 0.1596 |
| Qwen-Max | 0.4566 | 0.2676 | 0.1925 | 0.0871 | 0.1606 | 0.0761 | 0.0940 | 0.1248 | 0.0843 | 69.2308 | 3.5000 | 17.0200 | 73.9600 | 30.6700 | 49.0000 | 82.5500 | 0.2445 |
| Dolphin-V1 | 0.6819 | 0.5155 | 0.6943 | 0.5821 | 0.4775 | 0.6003 | 0.5080 | 0.2430 | 0.2273 | 38.6458 | 3.2193 | 15.1170 | 72.7287 | 54.0634 | 76.0111 | 92.9601 | 0.5835 |

DD = Disease Diagnosis; VRA = View Recognition and Assessment; LL = Lesion Localization; OD = Organ Detection; KD = Keypoint Detection; CVE = Clinical Value Estimation; RG = Report Generation; CG = Caption Generation.

## 4.4 LIMITATIONS AND FUTURE OUTLOOK FOR ULTRASOUND LVLMS

While existing LVLMs demonstrate impressive general multimodal capabilities, our results reveal fundamental limitations in ultrasound-specific perception and clinical reasoning.

**Weak Perception of Ultrasound Structures.** Models struggle with recognizing relative spatial relationships between anatomical structures, as reflected by their poor performance on detection tasks, and often fail to capture subtle echogenicity patterns that are essential for clinical diagnosis. This likely stems from the lack of large-scale ultrasound-specific image–caption pretraining data and the inherently noisy, heterogeneous nature of ultrasound imaging. Improving perception would require curated ultrasound datasets, ultrasound-aware pretraining objectives, and architectures or adapters with explicit spatial-reasoning capabilities.

**Clinical Ultrasound Tasks Are Far More Complex Than Generic Vision-Language Tasks.** Ultrasound spans more than 15 clinical subspecialties, each with distinct anatomical structures, scanning planes, and diagnostic criteria. For example, fetal biometry requires standardized abdominal circumference (AC) or head circumference (HC) views, while cardiac ultrasound relies on parasternal long-axis or apical four-chamber views. A clinically useful LVLM must therefore understand specialty-specific anatomy, follow established scanning protocols, and reason according to diagnostic workflows.

## 5 ANALYSIS

## 5.1 INSTRUCTION FOLLOWING ANALYSIS

Table 2 shows that contemporary models are already highly adept at parsing prompts and adhering to output specifications: six of the seventeen systems achieve a perfect score on the DD benchmark. The remaining models lag only slightly behind. The medical-oriented MiniGPT-Med (Alkhaldi et al., 2024) and MedDr (He et al., 2024) deliver middling results, while Qwen-3B and Qwen-72B (Bai et al., 2023b) close the gap rapidly as their parameter counts increase. Claude-3.7 (Anthropic, 2025) score of 0.942 is largely attributable to occasional formatting omissions. For every non-perfect model, the deviation from the maximum is under six percentage points, and no systematic failures are observed. We also note that some models occasionally refuse to answer due to internal safety constraints, producing responses such as "insufficient information" or "I cannot provide medical advice", rather than simply failing to follow instructions.

Table 2: Instruction following comparison across different models.

| Task | Models | | | | | | | | | | | | | | | | |
|------|--------------|-------|--------|--------|--------|--------|------------|----------|----------|-------|-------|---------|-----------|--------|---------|---------|-----------|
| | MiniGPT-Med | MedDr | Qwen-3B | Qwen-7B | Qwen-32B | Qwen-72B | Dolphin-V1 | DeepSeek | InternVL | LLaVA | Phi-4 | Mistral | Doubao-1.5 | GPT-4o | Gem-2.0 | Gem-2.5 | Claude-3.7 |
| DD | 0.952 | 0.961 | 0.968 | 0.983 | 0.996 | 1.000 | 1.000 | 1.000 | 0.993 | 0.987 | 0.998 | 0.999 | 1.000 | 1.000 | 0.997 | 1.000 | 0.942 |

## 5.2 PROMPT WITH OR WITHOUT ANATOMY

We investigate whether explicitly naming the anatomical region in the prompt significantly changes the diagnostic accuracy of LVLMs in ultrasound. To this end, we treat the two prompt variants as paired conditions applied to the same set of inputs and evaluate the statistical significance of their differences using McNemar's test.

Specifically, for each image $x_i$, we generate two prompts:

> **With anatomy**: "You are a radiologist analysing a {anatomy} ultrasound image, please analyze..."
> **No anatomy**: "You are a radiologist analysing an ultrasound image, please analyze..."

Each prompt–image pair is forward-passed through the model five times, with the final prediction determined by majority vote. This produces paired outcomes $(y_i^{\text{with}}, y_i^{\text{without}})$ for each image. Experiment was conduced on 521 breast and thyroid studies from our dataset, the following paired contingency table presents the result for model Gemini-2.0-Pro-Exp:

Table 3: **Effect of anatomy tokens in prompt design.** Paired outcomes of 521 samples comparing prompts with and without anatomy tokens. Each entry shows the number of samples in that outcome combination.

| With-anatomy prompt | No-anatomy prompt | |
|---------------------|-------------------|--------------------|
| | Correct | Incorrect |
| Correct | 209 *(both correct)* | 64 *(only anatomy correct)* |
| Incorrect | 26 *(only no-anatomy correct)* | 222 *(both incorrect)* |

McNemar's exact test yields a test statistic $\chi^2 = 16.04$ with $p = 6.2 \times 10^{-5}$, providing strong evidence that the two conditions differ. Specifically, prompts with anatomy tokens achieve an accuracy of 52.4% versus 45.1% without, a gain of +7.3 percentage points.

The McNemar test confirms that the inclusion of anatomy information in the prompt significantly improves diagnostic accuracy, rejecting the null hypothesis of no difference between prompt types.

## 6 CONCLUSION

Ultrasound is essential to global healthcare but remains difficult to interpret. We present **U2-BENCH**, the first benchmark for evaluating LVLMs on ultrasound understanding. It includes 7,241 cases across 15 anatomical regions and defines 8 clinical tasks for 50 application scenarios. Evaluating 20 LVLMs, we find their strong performance in classification but persistent challenges in spatial reasoning and clinical text generation, suggesting a future direction for improving LVLMs on ultrasound interpretation.

**Acknowledgements**   This research was conducted at Dolphin AI. We acknowledge support from the Hong Kong Research Grants Council (RGC) Early Career Scheme (Grant No. 22203525) and Fudan University.

**Reproducibility Statement**   We have taken extensive measures to ensure the reproducibility of our work. The benchmark dataset (7,241 ultrasound cases across 15 anatomies and 8 tasks) is publicly available on HuggingFace, and the complete evaluation toolkit is released anonymously at `https://anonymous.4open.science/r/U2-Bench-F781/VLMEVALKIT/`. Detailed descriptions of dataset curation, preprocessing, and quality verification procedures are provided in Section 3.2 and Appendix C–D, including sampling strategies, annotation protocols, and prompt templates. The full list of models evaluated, along with task-specific metrics and the aggregate U2-Score formulation, is given in Section 4.2 and Appendix C. For reproducibility of theoretical and statistical analyses (e.g., McNemar test for prompt design), contingency tables and test statistics are reported in Section 5.2.

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

## USAGE OF LLM

LLMs were used in this work as a writing and editing assistant. Specifically, they helped polish the language of the manuscript for clarity and conciseness, suggested alternative phrasings, and formatted some LaTeX tables. LLMs were not used for research ideation or experimental design, but were used to assist coding and prompt design.

## APPENDICES

Within this supplementary material, we elaborate on the following aspects:

- Appendix A: More Related Work and Future Work
- Appendix B: Safeguarding
- Appendix C: More Evaluation Details
- Appendix D: Prompt Details
- Appendix E: Dataset Details and License

## A LIMITATIONS AND FUTURE WORK

### A.1 MORE RELATED WORK

**Ultrasound Foundation Models.** Several ultrasound-specific models such as USFM (Jiao et al., 2024), UltraSAM (Meyer et al., 2024), and EchoFM (Kim et al., 2024) pretrain visual backbones using self-supervised or segmentation-oriented objectives. BiomedCLIP (Zhang et al., 2023), Fetal-CLIP (Maani et al., 2025), and EchoCLIP (Christensen et al., 2024) explore vision–language pretraining in biomedical domains, but are often narrow in scope (e.g., fetal or cardiac imaging only), require fine-tuning, and lack the general-purpose zero-shot capabilities of LVLMs. Our benchmark evaluates generalist LVLMs directly, across a diverse range of ultrasound tasks, without task-specific adaptation. Table 4 gives a detailed comparison of existing benchmarks and datasets.

Table 4: Comparison of U2-BENCH with existing benchmarks and foundation model datasets.

| Dataset / Benchmark | #Tasks | #Anatomies | #Val US Cases | Multimodal | Free-text Output | Public Available |
|---|---|---|---|---|---|---|
| **USFM** (Jiao et al., 2024) | 3 | 12 | 22,421 | ❌ | ❌ | ❌ |
| **UltraSam** (Meyer et al., 2024)[†] | 2 | 58 | 14,000 | ❌ | ❌ | ✅ |
| **FetalCLIP** (Maani et al., 2025) | 4 | Fetal | - | ✅ | ❌ | ❌ |
| **EchoCLIP** (Christensen et al., 2024) | 5 | Cardiac | 21,484 | ✅ | ✅ | ❌ |
| **EchoFM** (Kim et al., 2024) | 4 | Cardiac | - | ❌ | ❌ | ❌ |
| **GMAI-MMBench** (Chen et al., 2024a)[*] | 2 | 5 | ∼ 1,800 | ✅ | ❌ | ✅ |
| **U2-BENCH (Ours)** | 8 | 15 | 7,241 | ✅ | ✅ | ✅ |

[†]UltraSam's US-43d is composed of public datasets, but not released as a unified benchmark.
[*] We only count the statistics of the ultrasound part of the GMAI-MMBench dataset for a fair comparison.

### A.2 LIMITATIONS

**Ethical and Applicability Considerations.** U2-BENCH is designed as a research-oriented benchmark and is not intended for clinical deployment or diagnostic decision-making. Any real-world application of models evaluated on this benchmark would require separate validation and regulatory approval. Although all data sources are licensed or publicly available and de-identified where applicable, we acknowledge that not all ethical and demographic dimensions of fairness can be fully accounted for at this stage.

**Evaluation Scope.** The benchmark focuses on key task categories relevant to ultrasound interpretation—such as anatomical recognition, diagnostic classification, and structured report generation. While these tasks are representative and grounded in clinical utility, they do not exhaust the full landscape of sonographic applications. The evaluation metrics used (e.g., accuracy, BLEU) may not capture the full subtlety of expert clinical judgment, especially in edge cases.

**Ultrasound-Specific Challenges.** Ultrasound imaging is highly operator-dependent and subject to artifacts such as shadowing, speckle, and angle variation. Variability in scanning protocols and lack of standardized definitions (e.g., for "standard planes") can complicate model training and evaluation. These modality-specific challenges are inherent to ultrasound and reflect real-world complexities rather than flaws in the benchmark design.

### A.3 FUTURE WORK

**Extending Dataset Diversity and Robustness.** While U2-BENCH aggregates data from a broad range of sources, further expansion to include more institutions, device types, and global populations would improve its representativeness. Future iterations of the benchmark will explore domain adaptation, adversarial robustness, and performance under distribution shifts to better simulate deployment conditions in varied clinical environments.

**Model Generalization and Multimodal Reasoning.** Current LVLMs still struggle with fine-grained spatial tasks, consistency across subgroups, and robust generation of clinically meaningful

Table 5: **Summary of annotated datasets used in U2-BENCH, grouped by core capability and task.** The "Case Number" column indicates the number of samples per task, while "Total" reflects the overall count when available. More details about the datasets are included in Appendix E.

| Capability | Task | Case Number | Source Dataset | Total |
|---|---|---|---|---|
| Classification | DD | 1,411 | Breast Lesion Detection in Ultrasound Videos (Lin et al., 2022); Breast Ultrasound Images Dataset (Al-Dhabyani et al., 2020); Dermatologic Ultrasound Images for classification (Laverde Saad et al., 2021); Knee ultrasound dataset in a population-based cohort (Novin et al., 2023); KFGNet (NeuronXJTU & palkia1998, 2023); GDPHSYSUCC (Mo et al., 2022); LEPset (Li et al., 2023b); COVID-BLUES (Wiedemann et al., 2025); Ultrasound Guided Regional Anesthesia (Tyagi et al., 2024); Ultrasound Breast Images for Breast Cancer (Sairam, 2020); Algerian Ultrasound Images Thyroid Dataset: AUITD (Maroua, 2020); Auto-PCOS classification (Divekar & Sonawane, 2024) | 2,999 |
| | VRA | 1,588 | FETAL PLANES DB (Burgos-Artizzu et al., 2020); FPUS23 (Prabakaran et al., 2023); CAMUS (Leclerc et al., 2019); Knee ultrasound dataset in a population-based cohort (Novin et al., 2023); Thyroid (Krönke et al., 2022); ACOUSLIC-AI (Sappia, 2024); JNU-IFM (Lu et al., 2022); Carotid Artery Ultrasound and Color Doppler (Pahuni Choudhary, 2023); Auto-PCOS classification (Maroua, 2020); African Fetal Standard Plane (Sendra-Balcells et al., 2023); DDTI (Pedraza et al., 2015); CAMUS (Leclerc et al., 2019); CUBS (Meiburger et al., 2021); COVID-BLUES (Wiedemann et al., 2025); Dataset of B-mode fatty liver ultrasound images (Byra et al., 2018); The Open Kidney Ultrasound Dataset (Singla et al., 2023); Micro-Ultrasound Prostate Segmentation Dataset (Shao & Brisbane, 2024); Breast Ultrasound Images Dataset (Al-Dhabyani et al., 2020); Knee ultrasound dataset in a population-based cohort (Novin et al., 2023); Polycystic Ovary Ultrasound Images Dataset (Wisesty et al., 2018) | |
| Detection | LL | 503 | DDTI (Pedraza et al., 2015); Micro-Ultrasound Prostate Segmentation Dataset (Shao & Brisbane, 2024); Breast Ultrasound Images Dataset (Al-Dhabyani et al., 2020); KFGNet (NeuronXJTU & palkia1998, 2023); BrEaST (Pawłowska et al., 2024) | 2,921 |
| | OD | 1,918 | The Open Kidney Ultrasound Dataset (Singla et al., 2023); Echogenic (Da Correggio et al., 2023); FALLMUD (FALLMUD); CAMUS (Leclerc et al., 2019); HC18 (van den Heuvel et al., 2018); Thyroid (Krönke et al., 2022); CCA (Bi et al., 2024); Ultrasound Guided Regional Anesthesia (Tyagi et al., 2024); C-TRUS Dataset (Leenings et al., 2025); ACOUSLIC-AI (Sappia, 2024); PSFHS (Bai, 2024); JNU-IFM (Lu et al., 2022); US simulation & segmentation (Vitale et al., 2020) | |
| | KD | 500 | Unity Imaging Collaborative (Shun-Shin, 2023) | |
| Regression | CVE | 521 | CAMUS (Leclerc et al., 2019); CUBS (Meiburger et al., 2021); HC18 (van den Heuvel et al., 2018); ACOUSLIC-AI (Sappia, 2024); Dataset of B-mode fatty liver ultrasound images (Byra et al., 2018) | 521 |
| Generation | RG | 600 | Chinese Ultrasound Report Dataset (Li et al., 2024) | 800 |
| | CG | 200 | FPUS23 (Prabakaran et al., 2023) | |
| | | | **Overall Total** | **7,241** |

language. In future work, we aim to incorporate richer contextual information (e.g., patient history, multi-view inputs) to better assess models' multimodal integration capabilities and real-world reasoning performance.

**Video-Based and Real-Time Evaluation.** U2-BENCH currently operates on frame-based inputs to ensure comparability across models. However, clinical ultrasound interpretation often involves dynamic, probe-controlled acquisition. Extending the benchmark to include video sequences, real-time tasks, and longitudinal case studies will be a major step toward closing the simulation-to-clinic gap.

**Theoretical Foundations and Causality.** Our current benchmark is designed for practical performance evaluation. Future work will incorporate diagnostic reasoning audits, causal probing methods, and uncertainty quantification frameworks to deepen our understanding of LVLM behavior in high-stakes medical applications.

**Standardization in Ultrasound AI.** There is a growing need for community consensus on annotation standards, task definitions, and evaluation protocols in ultrasound AI. We hope U2-BENCH can serve as a starting point for these conversations and will actively evolve in response to feedback from both clinical and technical communities.

## B  SAFEGUARDING

This study involves secondary use of de-identified, publicly available or licensed ultrasound datasets for the purpose of benchmarking machine learning models. All data used in **U2-BENCH** are either publicly released with appropriate usage permissions or obtained through official licensing agreements. No personally identifiable information is used, and all experiments are conducted in accordance with relevant data protection and ethical guidelines. Human annotators involved in quality assurance were trained to follow data confidentiality protocols, and no clinical decision-making was involved at any stage of this work.

## C  MORE EVALUATION DETAILS

### C.1  DATASETS USED

In Table 5.

## C.2  Experiment Setting

We conducted experiments on **U2-Bench** with both open-source and closed-source LVLMs. Uniform prompts were applied across all models. The evaluation was executed on 32 NVIDIA A800 GPUs over a period of approximately two weeks, using the OpenCompass VLMEvalKit (Duan et al., 2024), with additional support from a unified framework (XiaohuMini, 2025). All models were tested with temperature 0.7.

**Evaluated Models.**    We evaluated 23 LVLMs, spanning both open-source and closed-source systems, and including both general-purpose and medical-specialized variants.

- **Qwen2.5-VL Series (Yang et al., 2024)**: This includes *Qwen2.5-VL-3B-Instruct*, *Qwen2.5-VL-7B-Instruct*, *Qwen2.5-VL-32B-Instruct*, *Qwen2.5-VL-72B-Instruct*
- **Medical-Specific Open-Source Models:** *MiniGPT-Med* (Wu et al., 2023b), *MedDr* (He et al., 2024), *Lingshu*, *MedGemma-4B* (Anil et al., 2023).
- **Other Open-Source Models:** *Phi-4-Multimodal-Instruct-5.6B* (Abdin et al., 2024), *InternVL3-9B-Instruct* (Zhu et al., 2025), *LLaVA-1.5-13B* (Liu et al., 2023a), *Mistral-Small-3.1-24B-Instruct-2503* (Jiang et al., 2023), *DeepSeek-VL2* (DeepSeek-AI, 2024)
- **Closed-Source Models:** *GPT-4o-Mini*, *GPT-4o-2024-08-06* (OpenAI, 2023), *GPT-5*, *Gemini-1.5-Pro (exp-02-05)*, *Gemini-2-Pro (exp-02-05)*, *Gemini-2.5-Pro-Preview (exp-02-05)* (Anil et al., 2023), *Claude-3-Sonnet (20250219)* (Anthropic, 2024), *Qwen-Max-2025-01-25* (Bai et al., 2023a), *Doubao-1.5-Vision-Pro-32K-250115* (ByteDance, 2024), *Dolphin-V1* (Model developed by *Dolphin AI*)
- **Random Guessing:** implemented by uniformly sampling from the valid answer set for each classification task.

## C.3  Justification of U2-Score Weighting

Table 6: **Task-specific evaluation metrics and weights.** The corresponding weight $w_t$ and metric used for overall score aggregation for each task.

| $t$ | 1
DD | 2
VRA | 3
LL | 4
OD | 5
KD | 6
CVE | 7
RG | 8
CG |
|---|---|---|---|---|---|---|---|---|
| $w_t$ | 0.2 | 0.2 | 0.07 | 0.27 | 0.07 | 0.07 | 0.08 | 0.04 |
| $d_t$ | Acc. | Acc. | Acc. | Acc. | Acc. | 1-RMSE | BLEU-4 | BLEU-4 |

The U2-Score summarizes model performance across the eight benchmark tasks in **U2-Bench** through a weighted aggregation:

$$\text{U2-Score} := \sum_{t=1}^{N} w_t \cdot d_t, \quad \text{where} \quad w_t = \frac{n_t}{\sum_j n_j}, \quad d_t \in [0, 1] \tag{2}$$

Each task $t$ is associated with a weight $w_t$ proportional to its number of annotated examples $n_t$, and a normalized evaluation score $d_t$ representing performance on that task. This formulation ensures that the final score reflects both task competence and dataset composition.

The weighting design of U2-Score is rooted in data-driven representation of ultrasound practice. All benchmark tasks are constructed from licensed and publicly available datasets sourced through a comprehensive and systematic search. As a result, the sample distribution across tasks captures the structure of real-world ultrasound data availability and usage. Tasks with greater sample counts typically correspond to more standardized, clinically widespread, and institutionally supported workflows—such as disease diagnosis and view recognition. Conversely, tasks with fewer examples often reflect more specialized applications or less frequently annotated modalities.

By aligning task weights with dataset size, the U2-Score encodes the practical significance and maturity of each task in the public ultrasound domain. This approach also promotes reliable aggregate evaluation, as scores are more robustly informed by tasks with greater data coverage. The resulting

metric balances comprehensiveness with grounded applicability, supporting fair and reproducible model comparison in the current landscape of ultrasound AI research.

## C.4 Error Analysis

### Diagnosis Task (DD)

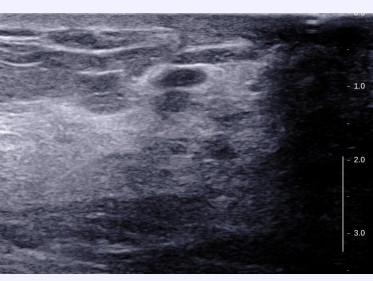

Figure 4: Ultrasound image for Diagnosis Task 40: case001273

**Accuracy**: 0.05 (1/20 models correct)

**Reference Answer**: 4A

**Model Responses**:

- **gemini-2.5-pro-preview-03-25**: 5
- **Do-3B**: 2
- **gpt-4o-2024-08-06**: '3'
- **doubao-1.5-vision-pro-32k-250115**: Insufficient information provided to determine BI - RADS category. Please describe the sonographic characteristics of the lesion in the image.
- **MedDr**: 5
- **qwen-max-2025-01-25**: 3
- **local_model**: 2
- **claude-3-7-sonnet-20250219**: 4C
- **gemini-2.0-pro-exp-02-05**: 4C
- **deepseek-vl2**: 4B
- **qwen2.5-vl-3b-instruct**: 2
- **Qwen2.5-VL-7B-Instruct-Pro**: 4A
- **gemini-1.5-pro-latest**: 2
- **Qwen2.5-VL-32B-Instruct**: 4B
- **InternVL3-9B-Instruct**: 4B
- **Phi-4-multimodal-instruct**: I cannot synthesize the sonographic characteristics of the lesions or make a BI-RADS assessment without the actual ultrasound images or a detailed description of the findings. If you provide the relevant details or images, I would be able to assist you with the assessment.
- **LLaVA-1.5-13B-HF**: '2'
- **Mistral-Small-3.1-24B-Instruct-2503**: 3
- **Do-7B**: The anatomical component most vividly portrayed is the breast.
- **gpt-4o-mini**: 2

**Prompt**: You are a radiologist analyzing a breast ultrasound image. Your task is to synthesize the sonographic characteristics of any identified lesions (or lack thereof) into a final ACR BI-RADS (Breast Imaging Reporting and Data System) assessment category.

BI-RADS Ultrasound Assessment Category Definitions

- '2' (Benign): Findings are definitively benign (e.g., simple cysts, intramammary lymph nodes, stable surgical implants/changes). 0% likelihood of malignancy. Requires routine screening follow-up.

- '3' (Probably Benign): Findings have characteristic benign features but are not definitively benign (e.g., presumed fibroadenoma, complicated cyst). Very low likelihood of malignancy (¿2%). Short-interval (e.g., 6-month) follow-up is typically recommended.

- '4A' (Low Suspicion for Malignancy): Findings warrant biopsy but have a low probability of malignancy (¿2% to ≤10%).

- '4B' (Moderate Suspicion for Malignancy): Findings warrant biopsy with an intermediate probability of malignancy (¿10% to ≤50%).

- '4C' (High Suspicion for Malignancy): Findings warrant biopsy with a high probability of malignancy (¿50% to ¡95%), without the classic features of Category 5.

- '5' (Highly Suggestive of Malignancy): Findings have classic malignant features (e.g., irregular spiculated mass). Very high probability of malignancy (≥95%). Biopsy is required, and definitive action should be taken regardless of pathology results if discordant.

Choose the single most appropriate BI-RADS assessment category from the options below.

options: ['2', '3', '4A', '4B', '4C', '5']

Output format: only the exact text of the chosen option from the list above. Do not include any introductory phrases, explanations, numbering, or formatting.

## View Recognition and Assessment Tasks (VRA)

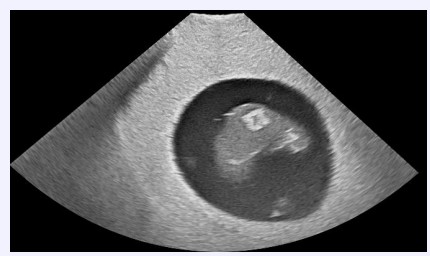

**Reference Answer**: hdvb

**Model Responses**:

- **Random Guessing**: huvf
- **MiniGPT-Med**: hdvf
- **MedDr**: hdvb
- **Qwen-2.5-VL-3B-Instruct**: hdvf
- **Qwen-2.5-VL-7B-Instruct**: hdvb
- **Qwen-2.5-VL-32B-Instruct**: hdvb
- **Qwen-2.5-VL-72B-Instruct**: hdvb
- **DeepSeek-VL2**: hdvf
- **InternVL3-9B-Instruct**: hdvf
- **LLaVA-1.5-13B**: huvb
- **Phi-4-Multimodal-Instruct**: hdvf
- **Mistral-Small-3.1-24B-Inst.**: hdvb
- **Doubao-1.5-Vision-Pro-32k**: hdvb
- **GPT-4o-Mini**: hdvf
- **GPT-4o**: hdvb
- **Gemini-1.5-Pro**: hdvf
- **Gemini-2.0-Pro-Exp**: hdvb
- **Gemini-2.5-Pro-Preview**: hdvf
- **Claude-3.7-Sonnet**: huvb
- **Qwen-Max**: hdvb
- **Dolphin-V1**: hdvb

*Prompt*: You are a radiologist analyzing a fetal ultrasound image. Your task is to determine the fetal presentation and orientation based on the provided ultrasound image. Specifically, identify whether the fetal head is down(hd) or up(hu). Additionally, determine if the fetal back is primarily oriented towards the ultrasound probe (vb) or towards the ultrasound probe (vf). Choose the single best option from the options below that accurately combines these findings. options: 'hdvb', 'hdvf', 'huvb', 'huvf' Output format: only the exact text of the chosen option from the list above. Do not include any introductory phrases, explanations, numbering, or formatting.

## Lesion Localization Tasks (LL)

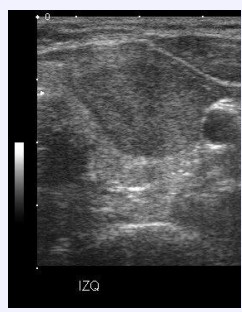

**Reference Answer**: upper left

**Model Responses**:

- **Random Guessing**: lower right
- **MiniGPT-Med**: upper center
- **MedDr**: upper left
- **Qwen-2.5-VL-3B-Instruct**: middle left
- **Qwen-2.5-VL-7B-Instruct**: upper left
- **Qwen-2.5-VL-32B-Instruct**: upper left
- **Qwen-2.5-VL-72B-Instruct**: upper left
- **DeepSeek-VL2**: upper right
- **InternVL3-9B-Instruct**: upper center
- **LLaVA-1.5-13B**: middle left
- **Phi-4-Multimodal-Instruct**: upper center
- **Mistral-Small-3.1-24B-Inst.**: upper left
- **Doubao-1.5-Vision-Pro-32k**: upper left
- **GPT-4o-Mini**: upper right
- **GPT-4o**: upper left
- **Gemini-1.5-Pro**: upper right
- **Gemini-2.0-Pro-Exp**: upper left
- **Gemini-2.5-Pro-Preview**: upper right
- **Claude-3.7-Sonnet**: upper center
- **Qwen-Max**: middle left
- **Dolphin-V1**: upper left

*Prompt*: You are a radiologist analyzing an ultrasound image of thyroid. Your task is to identify the primary location of any visible lesion(s) relative to the boundaries of the displayed image. Consider the lesion's center location or most prominent area when deciding. Choose the single option from the list below that best describes this location, even if the fit is approximate. options: upper left, upper center, upper right, middle left, center, middle right, lower left, lower center, lower right, not visible Output format: only the exact text of the chosen option from the list above. Do not include any introductory phrases, explanations, numbering, or formatting.

## Organ Detection Tasks (OD)

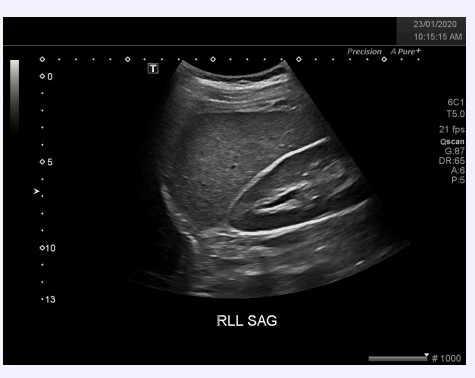

**Reference Answer**: center

**Model Responses**:

- **Random Guessing**: lower left
- **MiniGPT-Med**: middle right
- **MedDr**: center
- **Qwen-2.5-VL-3B-Instruct**: middle right
- **Qwen-2.5-VL-7B-Instruct**: center
- **Qwen-2.5-VL-32B-Instruct**: center
- **Qwen-2.5-VL-72B-Instruct**: center
- **DeepSeek-VL2**: middle left
- **InternVL3-9B-Instruct**: lower center
- **LLaVA-1.5-13B**: middle right
- **Phi-4-Multimodal-Instruct**: middle right
- **Mistral-Small-3.1-24B-Inst.**: center
- **Doubao-1.5-Vision-Pro-32k**: center
- **GPT-4o-Mini**: lower center
- **GPT-4o**: center
- **Gemini-1.5-Pro**: lower center
- **Gemini-2.0-Pro-Exp**: center
- **Gemini-2.5-Pro-Preview**: lower center
- **Claude-3.7-Sonnet**: center
- **Qwen-Max**: middle right
- **Dolphin-V1**: center

*Prompt*: You are a radiologist analyzing an ultrasound image of liver. Your task is to identify the primary location of the target organ relative to the boundaries of the displayed image. Consider the organ's center location or most prominent area when deciding. Choose the single option from the list below that best describes this location, even if the fit is approximate. options: upper left, upper center, upper right, middle left, center, middle right, lower left, lower center, lower right, not visible Output format: only the exact text of the chosen option from the list above. Do not include any introductory phrases, explanations, numbering, or formatting.

## Keypoint Detection Tasks (KD)

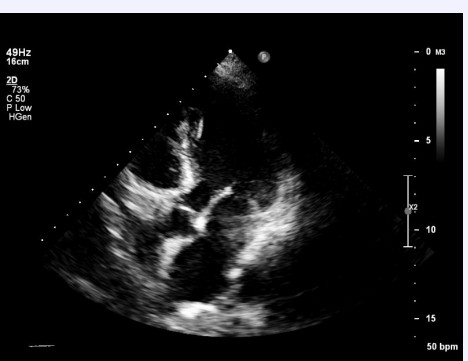

**Reference Answer**: middle right

**Model Responses**:

- **Random Guessing**: upper center
- **MiniGPT-Med**: middle left
- **MedDr**: middle right
- **Qwen-2.5-VL-3B-Instruct**: center
- **Qwen-2.5-VL-7B-Instruct**: middle right
- **Qwen-2.5-VL-32B-Instruct**: middle right
- **Qwen-2.5-VL-72B-Instruct**: middle right
- **DeepSeek-VL2**: center
- **InternVL3-9B-Instruct**: middle left
- **LLaVA-1.5-13B**: center
- **Phi-4-Multimodal-Instruct**: lower right
- **Mistral-Small-3.1-24B-Inst.**: middle right
- **Doubao-1.5-Vision-Pro-32k**: middle right
- **GPT-4o-Mini**: center
- **GPT-4o**: middle right
- **Gemini-1.5-Pro**: center
- **Gemini-2.0-Pro-Exp**: middle right
- **Gemini-2.5-Pro-Preview**: center
- **Claude-3.7-Sonnet**: middle right
- **Qwen-Max**: center
- **Dolphin-V1**: middle right

*Prompt*: You are a radiologist analyzing an ultrasound image of heart. Your task is to identify the primary location of the key anatomical landmark point relative to the boundaries of the displayed image. Consider the landmark's precise position when deciding. Choose the single option from the list below that best describes this location, even if the fit is approximate. options: upper left, upper center, upper right, middle left, center, middle right, lower left, lower center, lower right, not visible Output format: only the exact text of the chosen option from the list above. Do not include any introductory phrases, explanations, numbering, or formatting.

**Cardiac View Evaluation Tasks (CVE)**

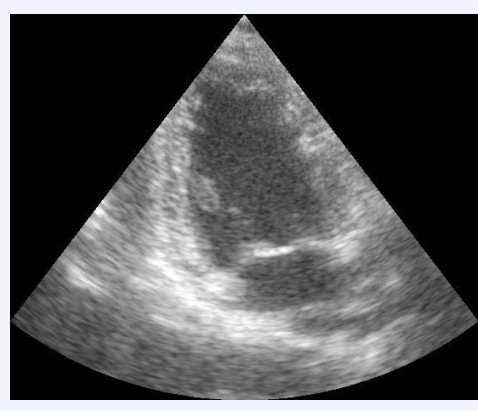

**Reference Answer**: 2CH

**Model Responses**:

- **Random Guessing**: 4CH
- **MiniGPT-Med**: 4CH
- **MedDr**: 2CH
- **Qwen-2.5-VL-3B-Instruct**: 4CH
- **Qwen-2.5-VL-7B-Instruct**: 4CH
- **Qwen-2.5-VL-32B-Instruct**: 4CH
- **Qwen-2.5-VL-72B-Instruct**: 4CH
- **DeepSeek-VL2**: 4CH
- **InternVL3-9B-Instruct**: 4CH
- **LLaVA-1.5-13B**: 4CH
- **Phi-4-Multimodal-Instruct**: 4CH
- **Mistral-Small-3.1-24B-Inst.**: 4CH
- **Doubao-1.5-Vision-Pro-32k**: 2CH
- **GPT-4o-Mini**: 4CH
- **GPT-4o**: 4CH
- **Gemini-1.5-Pro**: 4CH
- **Gemini-2.0-Pro-Exp**: 4CH
- **Gemini-2.5-Pro-Preview**: 4CH
- **Claude-3.7-Sonnet**: 2CH
- **Qwen-Max**: 4CH
- **Dolphin-V1**: 2CH

*Prompt*: You are a radiologist or cardiologist specializing in echocardiography, analyzing an apical view ultrasound image of the human heart.

Your task is to accurately identify the specific apical view presented in the provided echocardiogram image. Carefully examine the cardiac structures visible. Determine if the image displays primarily the left ventricle and left atrium only (indicative of a 2-Chamber view, 2CH), or if it clearly shows all four chambers: the left ventricle, right ventricle, left atrium, and right atrium (indicative of a 4-Chamber view, 4CH). Choose the single best option from the list below that correctly identifies the view.

options: 2CH, 4CH

Output format: only the exact text of the chosen option from the list above. Do not include any introductory phrases, explanations, numbering, or formatting.

## Cardiac Grading Tasks (CG)

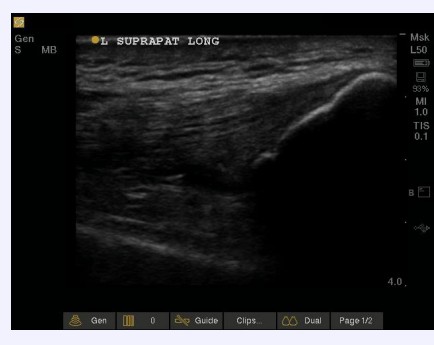

**Reference Answer**: Moderate OA

**Model Responses**:

- **Random Guessing**:
- **MiniGPT-Med**: Questionable OA
- **MedDr**: Moderate OA
- **Qwen-2.5-VL-3B-Instruct**: No OA
- **Qwen-2.5-VL-7B-Instruct**: Questionable OA
- **Qwen-2.5-VL-32B-Instruct**: Mild OA
- **Qwen-2.5-VL-72B-Instruct**: Mild OA
- **DeepSeek-VL2**: Questionable OA
- **InternVL3-9B-Instruct**: No OA
- **LLaVA-1.5-13B**: No OA
- **Phi-4-Multimodal-Instruct**: Questionable OA
- **Mistral-Small-3.1-24B-Inst.**: Mild OA
- **Doubao-1.5-Vision-Pro-32k**: Moderate OA
- **GPT-4o-Mini**: No OA
- **GPT-4o**: No OA
- **Gemini-1.5-Pro**: Questionable OA
- **Gemini-2.0-Pro-Exp**: Mild OA
- **Gemini-2.5-Pro-Preview**: Questionable OA
- **Claude-3.7-Sonnet**: Mild OA
- **Qwen-Max**: Mild OA
- **Dolphin-V1**: Moderate OA

*Prompt*: You are a radiologist analyzing an ultrasound image of left/right knee. Your task is to assess the severity of osteoarthritis (OA) using the established Kellgren-Lawrence (KL) grading system. Kellgren-Lawrence (KL) Grade Mapping to Options:
•'No OA': Corresponds to KL Grade 0 (No radiographic features of OA).
•'Questionable OA': Corresponds to KL Grade 1 (Doubtful JSN and possible minute osteophytes).
•'Mild OA': Corresponds to KL Grade 2 (Definite osteophytes and possible JSN).
•'Moderate OA': Corresponds to KL Grade 3 (Moderate multiple osteophytes, definite JSN, some sclerosis, possible deformity).
•'Severe OA': Corresponds to KL Grade 4 (Large osteophytes, marked JSN, severe sclerosis, definite deformity).
•'Total joint replacement': Indicates the presence of knee arthroplasty components (prosthesis), which replaces the native joint structures evaluated by the KL scale.
options: 'Mild OA', 'Moderate OA', 'No OA', 'Questionable OA', 'Severe OA', 'Total joint replacement'
Output format: only the exact text of the chosen option from the list above. Do not include any introductory phrases, explanations, numbering, or formatting.

**Report Generation Tasks (RG) Input**

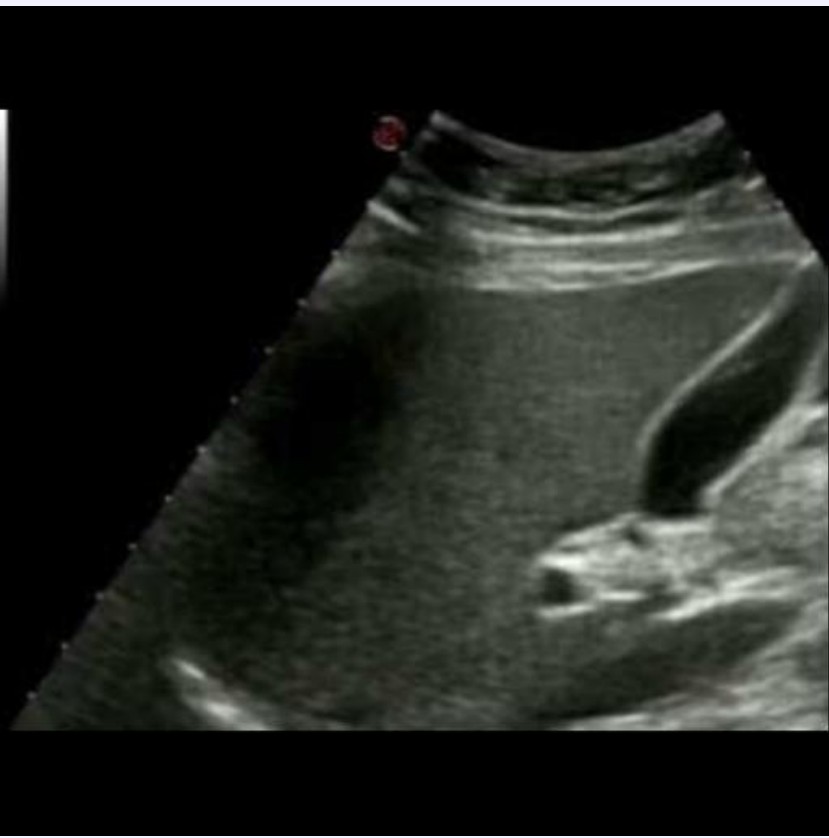

*Prompt*: You are a radiologist analyzing an ultrasound image focused on the Liver.
Your task is generate a concise and informative radiological report based strictly on the visual findings within the provided image. Your report should describe the primary organ's appearance (size, shape, borders/capsule), its parenchymal echotexture (e.g., homogeneous, heterogeneous, echogenicity relative to reference structures), and identify any visible abnormalities (e.g., masses, cysts, fluid collections, calcifications, ductal dilation). Comment on relevant adjacent structures if visualized. Use standard radiological terminology.
Output format: Strings, that is your report.
Example: The liver morphology is full with a smooth capsule. The parenchymal echotexture is fine and diffusely increased. Visualization of the portal venous system is suboptimal. Intrahepatic and extrahepatic bile ducts are not dilated. The main portal vein diameter is within normal limits. The gallbladder is normal in size and shape. The wall is smooth and not thickened. No obvious abnormal echoes are seen within the lumen. The pancreas is normal in size and shape with homogeneous parenchymal echotexture. The pancreatic duct is not dilated. No definite space-occupying lesion is seen within the pancreas. The spleen is normal in size and shape with homogeneous parenchymal echotexture. No obvious space-occupying lesion is seen within the spleen.

**Report Generation Tasks (RG) Output**

**Reference Answer**: The liver morphology is full with a smooth capsule. The parenchymal echotexture is fine and diffusely increased. Visualization of the portal venous system is suboptimal. Intrahepatic and extrahepatic bile ducts are not dilated. The main portal vein diameter is within normal limits. No focal lesions are identified.

**Model Responses**:

- **Random Guessing**:

- **MiniGPT-Med**: The liver shows normal size and contour with homogeneous echotexture. No focal lesions or biliary dilatation noted. Portal structures appear normal.

- **MedDr**: The liver demonstrates normal size and contour with a smooth capsule. The parenchymal echotexture is homogeneous with normal echogenicity. No focal lesions identified. Intrahepatic biliary ducts are not dilated. Portal venous structures appear normal.

- **Qwen-2.5-VL-3B-Instruct**: The liver appears normal in size with a smooth contour. The parenchymal echotexture is homogeneous without focal lesions. No intrahepatic biliary dilatation is seen.

- **Qwen-2.5-VL-7B-Instruct**: The liver appears normal in size with a smooth capsule. The parenchymal echotexture is homogeneous with no focal lesions. No intrahepatic or extrahepatic biliary dilatation is seen. The portal venous structures appear normal.

- **Qwen-2.5-VL-32B-Instruct**: The liver appears normal in size with a smooth contour. The parenchymal echotexture is homogeneous with normal echogenicity. No focal lesions such as masses, cysts, or calcifications are identified. The intrahepatic bile ducts are not dilated. The portal venous structures appear normal. No perihepatic fluid collection is observed.

- **Qwen-2.5-VL-72B-Instruct**: The liver demonstrates normal size and contour with a smooth capsule. The parenchymal echotexture is homogeneous with normal echogenicity. No focal lesions or masses are identified. The intrahepatic biliary tree is not dilated. Portal venous structures appear normal. No perihepatic fluid collection is seen.

- **DeepSeek-VL2**: The liver appears normal in size with a smooth contour. The parenchymal echotexture is homogeneous with normal echogenicity. No focal lesions are identified. The intrahepatic bile ducts are not dilated. The portal venous structures appear normal. No perihepatic fluid collection is observed.

- **InternVL3-9B-Instruct**: The liver appears normal in size with a smooth contour. The parenchymal echotexture is homogeneous without focal lesions. No intrahepatic or extrahepatic biliary dilatation is seen. The portal venous structures appear normal.

- **LLaVA-1.5-13B**: The liver appears normal in size with a smooth contour. The parenchymal echotexture is homogeneous without focal lesions. No intrahepatic biliary dilatation is seen. The portal structures appear normal.

- **Phi-4-Multimodal-Instruct**: The liver appears normal in size with a smooth contour. The parenchymal echotexture is homogeneous with no focal lesions. No intrahepatic or extrahepatic biliary dilatation is seen. The hepatic vessels appear normal.

- **Mistral-Small-3.1-24B-Inst.**: The liver demonstrates normal size and contour with a smooth capsule. The parenchymal echotexture is homogeneous with normal echogenicity. No focal lesions or masses are identified. The intrahepatic biliary tree is not dilated. Portal venous structures appear normal.

- **Doubao-1.5-Vision-Pro-32k**: The liver appears normal in size with a smooth contour. The parenchymal echotexture is homogeneous with normal echogenicity. No focal lesions are identified. The intrahepatic bile ducts are not dilated. The portal venous structures appear normal.

- **GPT-4o-Mini**: The liver appears normal in size with a smooth contour. The parenchymal echotexture is homogeneous without focal lesions. No intrahepatic or extrahepatic biliary dilatation is seen. The hepatic vessels appear normal. No ascites or other abnormalities are identified in the visualized portion of the liver.

- **GPT-4o**: The liver appears normal in size and shape with a smooth capsule. The parenchymal echotexture is homogeneous with no obvious focal lesions. No intrahepatic or extrahepatic bile duct dilation is noted. The gallbladder is not clearly visualized in this image. No ascites or other fluid collections are present. Adjacent structures are not well visualized for further evaluation.

- **Gemini-1.5-Pro**: The liver appears normal in size with a smooth contour. The parenchymal echotexture is homogeneous with normal echogenicity. No focal lesions such as masses, cysts, or calcifications are identified. The intrahepatic bile ducts are not dilated. The portal venous structures appear normal.

- **Gemini-2.0-Pro-Exp**: The liver appears normal in size with a smooth capsule. The parenchymal echotexture is homogeneous with normal echogenicity. No focal lesions are identified. The intrahepatic bile ducts are not dilated. The portal venous structures appear normal where visualized.

- **Gemini-2.5-Pro-Preview**: The liver appears normal in size with a smooth capsule. The parenchymal echotexture is homogeneous with normal echogenicity. No focal lesions such as masses, cysts, or calcifications are identified. The intrahepatic bile ducts are not dilated. The portal venous structures appear normal where visualized. No perihepatic fluid collection is seen. The gallbladder and other adjacent structures are not well visualized in this limited image. Overall impression: Normal liver parenchyma without evidence of focal lesions or biliary dilation.

- **Claude-3.7-Sonnet**: The liver demonstrates normal size and contour with a smooth capsule. The parenchymal echotexture appears homogeneous with mildly increased echogenicity, suggesting possible early fatty infiltration. No focal lesions, masses, or cysts are identified. The intrahepatic biliary tree is not dilated. Portal venous structures appear normal where visualized. No perihepatic fluid collection is seen. Adjacent structures are not well visualized in this limited view.

- **Qwen-Max**: The liver appears normal in size with a smooth contour. The parenchymal echotexture is homogeneous with normal echogenicity. No focal lesions such as masses, cysts, or calcifications are identified. The intrahepatic bile ducts are not dilated. The portal venous structures appear normal. No perihepatic fluid collection is observed. The visualized portions of adjacent structures appear unremarkable. Overall, the liver appears sonographically normal based on this limited view.

- **Dolphin-V1**: The liver morphology is full with a smooth capsule. The parenchymal echotexture is fine and diffusely increased. Visualization of the portal venous system is suboptimal. Intrahepatic and extrahepatic bile ducts are not dilated. The main portal vein diameter is within normal limits. No focal lesions are identified.

To provide deeper insights into model performance on medical imaging tasks, we conduct a comprehensive error analysis of models across four critical ultrasound image analysis tasks: measurement, classification, segmentation, and report generation. This analysis reveals distinct error patterns and task-specific challenges that inform future model improvements.

**Numerical Regression Task Analysis**   Among 101 total responses, the most significant challenge is the prevalence of **descriptive responses instead of numerical values** (53.47%). The model frequently generates interpretative text such as "The principal anatomical element visualized here is unequivocally the fetus head" rather than the expected numerical measurement (e.g., 291.4mm). This pattern suggests fundamental misunderstanding of task requirements, where the model interprets the task as image content identification rather than quantitative measurement.

Format violations constitute 1.98% of responses, where models provide numerical values with units (e.g., "113.6 mm") despite explicit formatting constraints. Notably, 43.56% of responses follow the correct numerical format, though accuracy assessment requires comparison with ground truth values. The high rate of descriptive responses indicates that current vision-language models struggle with the transition from visual analysis to precise quantitative output.

**Classification Task Performance**   Classification tasks demonstrate superior format compliance compared to measurement tasks, with 75.66% of responses providing valid option selections from 152 total responses. However, two distinct error patterns emerge: **explanatory responses** (5.92%) where models provide justifications rather than selections (e.g., "There is no definitive view of the fetal abdomen or pelvis to determine fetal position"), and **format violations** (18.42%) containing additional descriptive content alongside valid options.

The tendency toward explanatory responses reveals an interesting model behavior where excessive caution leads to task avoidance rather than best-effort selection from available options. This suggests that models may benefit from more explicit instructions emphasizing the requirement for definitive option selection even under uncertainty.

**Segmentation and Localization Analysis**   Segmentation tasks, requiring spatial reasoning for anatomical structure localization, show moderate success with 66% valid position responses from 500 total responses. The primary error categories include **invalid position terminology** (27.80%) with responses like "Not visible." or "Upper right." that contain punctuation or non-standard terms, and **complete task deviation** (6.20%) where models provide structural descriptions instead of positional information.

**Case Study Examples:** Analysis of specific segmentation cases reveals distinct model behaviors. In thyroid lesion localization tasks, while Gemini-2.5-Pro and GPT-4o consistently provide concise responses ("center"), Claude-3.7-Sonnet exhibits significant format violations. For instance, when tasked with identifying tumor location in breast ultrasound images, Claude generated extensive explanatory text:

> *"This image appears to be an ultrasound showing tissue layers with varying echogenicity... I cannot identify a clear, definitive lesion... For proper medical diagnosis, this ultrasound should be evaluated by a qualified radiologist..."*

Such responses, while demonstrating medical awareness, completely violate the specified output format requiring only location terms. This pattern suggests that Claude prioritizes safety disclaimers over task compliance in medical contexts.

Additionally, a concerning pattern emerges where multiple models consistently respond "center" regardless of actual lesion position, as evidenced by reference bounding boxes indicating lesions at coordinates [0.6, 0.247] and [0.595, 0.308]. This suggests potential spatial reasoning limitations or default response bias that could compromise clinical utility.

The relatively high success rate in spatial localization compared to numerical measurement suggests that discrete spatial reasoning may be more accessible to current vision-language architectures than continuous numerical estimation.

**Report Generation Excellence**    Report generation tasks achieve the highest success rate (98%) among all evaluated tasks, with only 2% exhibiting structural misidentification and 1% showing false findings. The rare but critical errors include anatomical misidentification ("Top view of fetus head and thorax" for fetal head ultrasound) and false pathological findings ("Aneuploid fetus with abnormal facial features"). While infrequent, such errors carry significant clinical implications, potentially leading to unnecessary medical interventions or patient anxiety.

**Cross-Task Error Pattern Analysis**    Task difficulty ranking from most to least challenging reveals: measurement (43.56% success) > segmentation (66% success) > classification (75.66% success) > report generation (98% success). This hierarchy reflects the increasing complexity of transitioning from free-form text generation to structured, constrained outputs requiring precise adherence to format specifications.

Common error patterns across tasks include: (1) **descriptive language substitution**, most prominent in measurement tasks where models default to interpretative text rather than required numerical values; (2) **format non-compliance**, prevalent across classification and segmentation tasks despite clear formatting instructions; and (3) **task misunderstanding**, where models completely misinterpret task objectives, such as treating localization as structure identification.

**Implications for Medical AI Development**    These findings highlight critical considerations for deploying vision-language models in medical imaging applications. The inverse relationship between task constraint and model performance suggests that current architectures excel at unconstrained text generation but struggle with precise, structured outputs essential for clinical decision-making. Future developments should prioritize: (1) enhanced instruction following capabilities for constrained output generation, (2) domain-specific fine-tuning on medical imaging tasks emphasizing numerical precision, and (3) robust validation mechanisms to detect and prevent false findings in clinical applications.

The analysis underscores that while large vision-language models show promise for medical imaging applications, careful task-specific optimization and human oversight remain essential, particularly for quantitative measurements and diagnostic assessments where precision directly impacts patient care.

## D  PROMPT FOR TASKS

---

**Prompt Template used for fetal view classification (dataset 10)**

You are a radiologist analyzing a fetal ultrasound image.

Your task is to determine the fetal presentation and orientation based on the provided ultrasound image. Specifically, identify whether the fetal head is down(hd) or up(hu). Additionally, determine if the fetal back is primarily oriented towards the ultrasound probe (vb) or towards the ultrasound probe (vf). Choose the single best option from the options below that accurately combines these findings.

options: 'hdvb', 'hdvf', 'huvb', 'huvf'

Output format: only the exact text of the chosen option from the list above. Do not include any introductory phrases, explanations, numbering, or formatting.

---

**Prompt Template used for heart view classification (dataset 18)**

You are a radiologist or cardiologist specializing in echocardiography, analyzing an apical view ultrasound image of the human heart.

Your task is to accurately identify the specific apical view presented in the provided echocardiogram image. Carefully examine the cardiac structures visible. Determine if the image displays primarily the left ventricle and left atrium only (indicative of a 2-Chamber view, 2CH), or if it clearly shows all four chambers: the left ventricle, right ventricle, left atrium, and right atrium (indicative of a 4-Chamber view, 4CH). Choose the single best option from the list below that correctly identifies the view.

options: 2CH, 4CH

Output format: only the exact text of the chosen option from the list above. Do not include any introductory phrases, explanations, numbering, or formatting.

---

**Prompt Template used for (KL) grading (dataset 28)**

You are a radiologist analyzing an ultrasound image of left/right knee.

Your task is to assess the severity of osteoarthritis (OA) using the established Kellgren-Lawrence (KL) grading system. Kellgren-Lawrence (KL) Grade Mapping to Options:

- 'No OA': Corresponds to KL Grade 0 (No radiographic features of OA).
- 'Questionable OA': Corresponds to KL Grade 1 (Doubtful JSN and possible minute osteophytes).
- 'Mild OA': Corresponds to KL Grade 2 (Definite osteophytes and possible JSN).
- 'Moderate OA': Corresponds to KL Grade 3 (Moderate multiple osteophytes, definite JSN, some sclerosis, possible deformity).
- 'Severe OA': Corresponds to KL Grade 4 (Large osteophytes, marked JSN, severe sclerosis, definite deformity).
- 'Total joint replacement': Indicates the presence of knee arthroplasty components (prosthesis), which replaces the native joint structures evaluated by the KL scale.

Choose the single best option from the following list that accurately describes the image.

options: 'Mild OA', 'Moderate OA', 'No OA', 'Questionable OA', 'Severe OA', 'Total joint replacement'

Output format: only the exact text of the chosen option from the list above. Do not include any introductory phrases, explanations, numbering, or formatting.

---

**Prompt Template used for BI-RADS classification (dataset 40)**

You are a radiologist analyzing a breast ultrasound image.

Your task is to synthesize the sonographic characteristics of any identified lesions (or lack thereof) into a final ACR BI-RADS (Breast Imaging Reporting and Data System) assessment category.

BI-RADS Ultrasound Assessment Category Definitions:
- '2' (Benign): Findings are definitively benign (e.g., simple cysts, intramammary lymph nodes, stable surgical implants/changes). 0% likelihood of malignancy. Requires routine screening follow-up.
- '3' (Probably Benign): Findings have characteristic benign features but are not definitively benign (e.g., presumed fibroadenoma, complicated cyst). Very low likelihood of malignancy (¡2%). Short-interval (e.g., 6-month) follow-up is typically recommended.
- '4A' (Low Suspicion for Malignancy): Findings warrant biopsy but have a low probability of malignancy (¿2% to $\leq$10%).
- '4B' (Moderate Suspicion for Malignancy): Findings warrant biopsy with an intermediate probability of malignancy (¿10% to $\geq$50%).
- '4C' (High Suspicion for Malignancy): Findings warrant biopsy with a high probability of malignancy (¿50% to ¡95%), without the classic features of Category 5.
- '5' (Highly Suggestive of Malignancy): Findings have classic malignant features (e.g., irregular spiculated mass). Very high probability of malignancy ($\geq$95%). Biopsy is required, and definitive action should be taken regardless of pathology results if discordant.

Choose the single most appropriate BI-RADS assessment category from the options below.

options: ['2', '3', '4A', '4B', '4C', '5']

Output format: only the exact text of the chosen option from the list above. Do not include any introductory phrases, explanations, numbering, or formatting.

---

**Prompt Template used for fetal abdomen (dataset 50)**

You are a radiologist analyzing an ultrasound image of fetal abdomen.

Your task is to determine if the presented cross-sectional view of the fetal abdomen is technically adequate for performing an accurate Abdominal Circumference (AC) measurement according to standard obstetric guidelines. Identify the specific anatomical plane shown for the fetal abdomen. Determine if this plane meets the criteria for an optimal AC measurement (correct landmarks visible, proper transverse orientation) or if it is suboptimal (incorrect plane, missing landmarks, oblique/foreshortened view, presence of interfering structures like kidneys). Choose the single best option describing the plane's suitability for AC measurement.

options: 'none', 'optimal', 'suboptimal'

Output format: only the exact text of the chosen option from the list above. Do not include any introductory phrases, explanations, numbering, or formatting.

---

**Prompt Template used for breast classification**

You are a radiologist analyzing a breast ultrasound image.

Your task is carefully examine the provided breast ultrasound image, evaluate any identified lesions or abnormalities based on key sonographic characteristics (including shape, orientation, margin, echo pattern, posterior acoustic features, and associated features), synthesize these features to form an overall impression about the likelihood of malignancy, and then choose the single best option from the following list that accurately summarizes this assessment.

options: (normal), benign, malignant

Output format: only the exact text of the chosen option from the list above. Do not include any introductory phrases, explanations, numbering, or formatting.

---

**Prompt Template used for thyroid classification**

You are a radiologist specializing in head and neck or endocrine imaging, analyzing an ultrasound image of the thyroid gland.

Your task is to carefully examine the provided thyroid ultrasound image, evaluate the overall thyroid gland parenchyma (echogenicity, texture, vascularity), identify any focal nodules, assess the specific sonographic features of any nodules found (including composition, echogenicity, shape, margin, and echogenic foci), synthesize these findings to determine if the gland appears normal, contains benign-appearing findings, or contains findings suspicious for malignancy, and then choose the single best option from the following list that accurately summarizes this assessment.

options: (normal thyroid), benign, malignant

Output format: only the exact text of the chosen option from the list above. Do not include any introductory phrases, explanations, numbering, or formatting.

---

**Prompt Template used for skin cancer classification (dataset 25)**

You are a radiologist analyzing an ultrasound image of skin.

Your task is to carefully examine the provided skin ultrasound image, evaluate the identified lesion or abnormality based on key sonographic characteristics (including its location within skin layers, echogenicity, internal echo texture, shape, margins, size/depth, posterior acoustic phenomena, and vascularity assessed with Doppler), synthesize these features to form an overall impression regarding the likelihood of malignancy, and then choose the single best option from the following list that summarizes this assessment.

options: benign, malignant

Output format: only the exact text of the chosen option from the list above. Do not include any introductory phrases, explanations, numbering, or formatting.

---

**Prompt Template used for pancreas cancer classification (dataset 42)**

You are a radiologist analyzing an ultrasound image of the pancreas.

Your task is to carefully examine the provided ultrasound image of the pancreas, evaluate the gland's echotexture, size, margins, and the pancreatic duct diameter, identify any focal lesions or masses (noting their echogenicity, margins, size, and vascularity if Doppler is available), assess for associated findings such as ductal dilation (including potential "double duct" sign), vascular involvement (encasement/thrombosis), regional lymphadenopathy, or fluid collections, synthesize these findings to determine if there is evidence suspicious for primary pancreatic cancer versus other findings, and then choose the single best option from the following list that summarizes this assessment.

options: non-pancreas cancer, pancreas cancer

Output format: only the exact text of the chosen option from the list above. Do not include any introductory phrases, explanations, numbering, or formatting.

---

**Prompt Template used for PCOS classification (dataset 74)**

You are a radiologist analyzing an ultrasound image obtained during a pelvic examination, potentially as part of an evaluation for Polycystic Ovary Syndrome (PCOS).

Your task is to evaluate the overall appearance of the anatomical structures presented in the ultrasound image (primarily focusing on the ovaries and potentially the uterus). Consider sonographic features such as ovarian size, morphology, follicle count and distribution, stromal echogenicity, as well as any other findings that might indicate pathology. Based on this assessment, determine if the image appears generally normal or if it displays features suggestive of an abnormality (which could include findings consistent with PCOS or other conditions). Choose the single best option from the following list that accurately describes this overall impression.

options: 'Appears abnormal', 'Appears normal'

Output format: only the exact text of the chosen option from the list above. Do not include any introductory phrases, explanations, numbering, or formatting.

---

**Prompt Template used for PCOS classification (dataset 74)**

You are a radiologist analyzing an ultrasound image obtained during a pelvic examination. Crucially, assume this specific image has already been determined to show some form of abnormality. Your focus now is on the nature of that abnormality.

Your task is to specifically assess whether the abnormality present in this ultrasound image includes clear sonographic evidence consistent with a polycystic ovary. Evaluate the visualized ovarian structures, paying close attention to features commonly associated with PCOS, such as: increased number of follicles, peripheral distribution of follicles, increased ovarian volume, increased stromal echogenicity or volume. Based on whether these specific PCOS-related sonographic features are identifiable within the overall abnormal appearance, specifies whether the ultrasound image shows evidence/ visibility of a polycystic ovary or not. Choose the single best option from the following list.

options: 'Not-visible', 'Visible'

Output format: only the exact text of the chosen option from the list above. Do not include any introductory phrases, explanations, numbering, or formatting.

---

**Prompt Template used for PCOS classification (dataset 75)**

You are a radiologist analyzing an ultrasound image obtained during a pelvic examination, specifically being evaluated for features potentially related to Polycystic Ovary Syndrome (PCOS).

Your task is to carefully evaluate the provided ultrasound image for sonographic features consistent with Polycystic Ovarian Morphology (PCOM), which is the ultrasound component relevant to PCOS detection. Analyze the visualized ovary (or ovaries), considering criteria such as increased ovarian volume, increased antral follicle count (e.g., $\geq 20$ per ovary), peripheral follicle distribution, and / or increased stromal echogenicity / volume. If sonographic features consistent with PCOM are present, select the label 'infected', otherwise 'noninfected'. Choose the single best option from the following list.

options: 'infected', 'noninfected'

Output format: only the exact text of the chosen option from the list above. Do not include any introductory phrases, explanations, numbering, or formatting.

---

**Prompt Template used for lung parenchyma (dataset 44)**

You are a radiologist or clinician skilled in performing and interpreting Lung Ultrasound (LUS), specifically analyzing an ultrasound image of the lung pleura and parenchyma.

Your task is to carefully examine the provided lung ultrasound image, focusing on the appearance of the pleural line and the underlying lung parenchyma, identify the presence and characteristics of A-lines, B-lines (number, coalescence), and any consolidations according to the defined severity scoring criteria below, and then choose the single best integer score (0, 1, 2, or 3) from the following list that accurately reflects the observed findings.

LUS Severity Score Criteria:
- 0: Normal lung pattern. Characterized by a continuous, regular, thin pleural line with horizontal reverberation artifacts (A-lines) below it. Sliding lung sign is typically present.
- 1: Mild interstitial syndrome. Characterized by an indented or slightly irregular pleural line. Scattered, well-defined vertical artifacts (B-lines) are visible (typically $\geq 3$ B-lines per intercostal space but not coalescent).
- 2: Moderate interstitial syndrome or early consolidation. Characterized by a broken or significantly irregular pleural line. Multiple coalescent B-lines (small "white lung" areas) or small subpleural consolidations are present.
- 3: Severe interstitial syndrome or large consolidation. Characterized by dense and largely extended confluent B-lines ("white lung" appearance occupying most or all of the screen) with or without large consolidations.

Options: 0, 1, 2, 3

Output format: only the single chosen integer number from the list above. Do not include any introductory phrases, explanations, numbering, or formatting.

Prompt Template used for fatty liver classification (dataset 57)

You are a radiologist analyzing a static B-mode ultrasound image displaying the liver.

Your task is to evaluate the liver parenchyma in the provided image to determine the grade of hepatic steatosis. For this task, label 1 is assigned if the image displays features consistent with fatty liver (which often correlates histologically with ¿5% hepatocyte steatosis), while label 0 is assigned if such features are absent. Based on your comprehensive assessment of these sonographic features, determine whether the image displays sufficient evidence to be classified as showing fatty liver (Label 1) or not (Label 0). Choose the single best option from the following list that accurately reflects your classification.

options: 0, 1

Output format: only the single chosen integer number from the list above. Do not include any introductory phrases, explanations, numbering, or formatting.

Prompt Template used for fetal (dataset 03)

You are a radiologist analyzing a single ultrasound image acquired during a fetal examination.

Your task is to carefully examine the provided image, identify the primary anatomical structure or region being visualized, and determine the most appropriate description based on the standard imaging planes used in fetal ultrasound. Choose the single best option from the following list that accurately describes the main subject shown in the image.

options: 'fetal abdomen','fetal femur','fetal brain', 'fetal thorax', 'maternal cervix', 'other'

Output format: only the exact text of the chosen option from the list above. Do not include any introductory phrases, explanations, numbering, or formatting.

Prompt Template used for throid plane classification (dataset 37)

You are a radiologist with expertise in interpreting neck and thyroid ultrasound images. You are presented with a single B-mode ultrasound image focused on the thyroid gland and adjacent neck structures.

Your task is to identify the Cardinal Anatomical Plane depicted in the provided ultrasound image. Choose the single best option from the following list that accurately describes the image.

options: 'Axial/Transverse Plane', 'Coronal Plane', 'Sagittal Plane'

Output format: only the exact text of the chosen option from the list above. Do not include any introductory phrases, explanations, numbering, or formatting.

---

**Prompt Template used for fetal (dataset 53)**

You are a radiologist analyzing a single B-mode ultrasound image obtained during a fetal assessment.

Your task is to carefully examine the provided ultrasound image frame to identify the presence or absence of two specific anatomical landmarks: the fetal head and the maternal symphysis pubis. Based on this identification, classify the frame's content by choosing the single best option from the following list that accurately describes which of these landmarks are visible. Choose the single best option from the following list that accurately describes the frame's content.

options: 'None', 'OnlyFetalHead', 'OnlySymphysisPubis', 'SymphysisPubis+FetalHead'

Output prompt: only the exact text of the chosen option from the list above. Do not include any introductory phrases, explanations, numbering, or other formatting.

---

**Prompt Template used for cartoid classification (dataset 69)**

You are a radiologist analyzing an ultrasound image depicting a portion of the carotid arterial system in the neck.

Your task is to carefully examine the provided ultrasound image, analyzing anatomical landmarks, vessel morphology, and its position relative to other neck structures, to identify the primary carotid artery segment shown. Choose the single best option from the following list that accurately describes the main vessel visualized in the frame's content. Assume 'left carotid' and 'right carotid' refer generally to the common or internal carotid artery on that respective side, while 'external carotid' refers specifically to the external carotid artery branch. Choose the single best option from the following list that accurately describes the image.

options: 'external carotid', 'left carotid', 'right carotid'

Output prompt: only the exact text of the chosen option from the list above. Do not include any introductory phrases, explanations, numbering, or other formatting.

---

**Prompt Template used for anatomy classification**

You are an expert specialized in analyzing medical ultrasound images. You are provided with a single ultrasound image frame, which could depict various parts of the human body.

Your task is to analyze the provided ultrasound image and identify the primary anatomical region or organ system being visualized. Choose the single best option from the following list that most accurately represents this primary anatomical subject.

options: 'fetal', 'thyroid', 'heart', 'lung', 'liver', 'carotid', 'kidney', 'prostate', 'breast', 'other'

Output prompt: only the exact text of the chosen option from the list above. Do not include any introductory phrases, explanations, numbering, or other formatting.

---

**Prompt Template used for knee classification**

You are a radiologist analyzing an ultrasound image of knee.

Your task is to classify the specific anatomical view, laterality (left/right), orientation, and any specific imaging technique or patient positioning shown in the image:

- 'left anterior suprapatellar longitudinal': Image of the left knee, taken from the front (anterior), just above the kneecap (suprapatellar), with the ultrasound probe oriented along the long axis of the thigh/patellar tendon. Standard B-mode imaging.
- 'left anterior suprapatellar longitudinal with power Doppler': Same view as above (left, anterior suprapatellar, longitudinal), but with Power Doppler mode activated, typically used to assess blood flow or inflammation.
- 'left anterior suprapatellar transverse in 30 degrees flexion': Image of the left knee, from the front (anterior), above the kneecap (suprapatellar), with the probe oriented across (transverse) the thigh, and the knee bent at approximately 30 degrees.
- 'left anterior suprapatellar transverse in maximal flexion': Same view as above (left, anterior suprapatellar, transverse), but with the knee bent as much as possible (maximal flexion).
- 'left lateral longitudinal': Image of the outer side (lateral) of the left knee, with the probe oriented along the long axis of the structures (e.g., LCL, IT band).
- 'left medial longitudinal': Image of the inner side (medial) of the left knee, with the probe oriented along the long axis of the structures (e.g., MCL, medial meniscus).
- 'left posterior medial transverse': Image of the back, inner corner (posterior medial) of the left knee, with the probe oriented across (transverse) the structures (often used for Baker's cysts).
- 'right anterior suprapatellar longitudinal': Image of the right knee, taken from the front (anterior), just above the kneecap (suprapatellar), with the ultrasound probe oriented along the long axis of the thigh/patellar tendon. Standard B-mode imaging.
- 'right anterior suprapatellar longitudinal with power Doppler': Same view as above (right, anterior suprapatellar, longitudinal), but with Power Doppler mode activated.
- 'right anterior suprapatellar transverse in 30 degrees flexion': Image of the right knee, from the front (anterior), above the kneecap (suprapatellar), with the probe oriented across (transverse) the thigh, and the knee bent at approximately 30 degrees.
- 'right anterior suprapatellar transverse in maximal flexion': Same view as above (right, anterior suprapatellar, transverse), but with the knee bent as much as possible (maximal flexion).
- 'right lateral longitudinal': Image of the outer side (lateral) of the right knee, with the probe oriented along the long axis of the structures.
- 'right medial longitudinal': Image of the inner side (medial) of the right knee, with the probe oriented along the long axis of the structures.
- 'right posterior medial transverse': Image of the back, inner corner (posterior medial) of the right knee, with the probe oriented across (transverse) the structures.

Choose the single best option from the following list that accurately describes the image.

Options: 'left anterior suprapatellar longitudinal', 'left anterior suprapatellar longitudinal with power Doppler', 'left anterior suprapatellar transverse in 30 degrees flexion', 'left anterior suprapatellar transverse in maximal flexion', 'left lateral longitudinal', 'left medial longitudinal', 'left posterior medial transverse', 'right anterior suprapatellar longitudinal', 'right anterior suprapatellar longitudinal with power Doppler', 'right anterior suprapatellar transverse in 30 degrees flexion', 'right anterior suprapatellar transverse in maximal flexion', 'right lateral longitudinal', 'right medial longitudinal', 'right posterior medial transverse'

Output prompt: only the exact text of the chosen option from the list above. Do not include any introductory phrases, explanations, numbering, or other formatting.

---

**Prompt Template used for lesion detection**

You are a radiologist analyzing an ultrasound image of thyroid.

Your task is to identify the primary location of any visible lesion(s) relative to the boundaries of the displayed image. Consider the lesion's center location or most prominent area when deciding. Choose the single option from the list below that best describes this location, even if the fit is approximate.

Choose the single most appropriate location from the following list:
- upper left
- upper center
- upper right
- middle left
- center
- middle right
- lower left
- lower center
- lower right
- not visible

Output format: only one or two word(s) representing the chosen location. No additional text or formatting is allowed.

---

**Prompt Template used for organ detection**

You are a radiologist analyzing an ultrasound image of abdominal.

Your task is to determine the primary location, relative to the image boundaries, for each visible structure listed in liver.

- Consider the structure's center or most prominent area when deciding its location.
- Choose the single option from the list below that best describes the location, even if the fit is approximate.

Location Options:
- upper left
- upper center
- upper right
- middle left
- center
- middle right
- lower left
- lower center
- lower right
- not visible

Output format: only one or two word(s) representing the chosen location. No additional text or formatting is allowed.

**Prompt Template used for keypoint detection**

You are a radiologist analyzing an ultrasound image of the heart.

Your task is to determine the top inner point of the aortic valve.
- Consider the structure's center or most prominent area when deciding its location.
- Choose the single option from the list below that best describes the location, even if the fit is approximate.

Location Options:
- upper left
- upper center
- upper right
- middle left
- center
- middle right
- lower left
- lower center
- lower right
- not visible

Output format: only one or two word(s) representing the chosen location. No additional text or formatting is allowed.

**Prompt Template used for caption generation**

You are a radiologist analyzing an ultrasound image focused on the {anatomy_location}.

Your task is to generate a concise and informative caption that accurately describes the key anatomical structures and any significant findings visible in the provided ultrasound image.

Output format: A single string constituting the image caption. Output only the generated caption text itself. Do not include any introductory phrases (like Ċaption:̈), labels, explanations, or additional formatting.

Examples:
Example1: Thyroid nodule in the right lobe. TI-RADS level 3, Benign.
Example2: Thyroid nodule in the left lobe. TI-RADS level 3, Benign.
Example3: Thyroid nodule in the right lobe. TI-RADS level 4, Benign.

> **Prompt Template used for report generation**
>
> You are a radiologist analyzing an ultrasound image focused on the {anatomy_location}.
>
> Your task is generate a concise and informative radiological report based strictly on the visual findings within the provided image. Your report should describe the primary organ's appearance (size, shape, borders/capsule), its parenchymal echotexture (e.g., homogeneous, heterogeneous, echogenicity relative to reference structures), and identify any visible abnormalities (e.g., masses, cysts, fluid collections, calcifications, ductal dilation). Comment on relevant adjacent structures if visualized. Use standard radiological terminology.
>
> Output format: Strings, that is your report.
>
> Example: The liver morphology is full with a smooth capsule. The parenchymal echotexture is fine and diffusely increased. Visualization of the portal venous system is suboptimal. Intrahepatic and extrahepatic bile ducts are not dilated. The main portal vein diameter is within normal limits. The gallbladder is normal in size and shape. The wall is smooth and not thickened. No obvious abnormal echoes are seen within the lumen. The pancreas is normal in size and shape with homogeneous parenchymal echotexture. The pancreatic duct is not dilated. No definite space-occupying lesion is seen within the pancreas. The spleen is normal in size and shape with homogeneous parenchymal echotexture. No obvious space-occupying lesion is seen within the spleen.

# E  DATASET DETAILS AND LICENSE

Table 7: Summary of Annotated Datasets Used in U2-BENCH. In particular, 25 out of the 40 datasets were initially identified and compiled by Alsharid et al. (2025), and the corresponding entries here were adapted from the tables in that work. For a more comprehensive dataset-level characterization of that data, we refer readers to that paper.

| Dataset | Anatomy | Clinical scenarios | Task | Case | License |
|---|---|---|---|---|---|
| FETAL PLANES DB (Burgos-Artizzu et al., 2020) | Fetal abdomen Fetal brain Fetal femur Fetal thorax Maternal cervix other | Fetal standard plane identification | VRA | 137 | CCA 4.0I |
| DDTI (Pedraza et al., 2015) | thyroid | Thyroid nodule identification Thyroid nodule localisation | VRA LL | 110 | - |
| The Open Kidney US Dataset (Singla et al., 2023) | kidney | Kidney detection Kidney Diag view identification | VRA OD | 110 | CC BY-NC-SA |
| FPUS23 (Prabakaran■ et al., 2023) | Fetal abdomen Fetal arm Fetal head Fetal legs | Fetal diagnostic planes identification Fetal US report generation | VRA RP | 752 | MIT |
| Echogenic (Da Correggio et al., 2023) | Fetal abdomen | Fetal abdominal organ detection | OD | 102 | CCA 4.0 |
| FALLMUD (FALL-MUD) | Crural muscles | Muscle detection | OD | 100 | - |
| Micro-US Prostate Segmentation Dataset (Shao & Brisbane, 2024) | Prostate | Prostate localisation Prostate Diag view identification | VRA LL | 110 | CCA 4.0I |
| CAMUS (Leclerc et al., 2019) | Heart ED Heart ES Heart 2CH Heart 4CH | Heart ejection fraction estimation Heart atrium and ventricle localisation | VRA OD CVE | 316 | CC BY-NC-SA 4.0 |
| Breast Lesion Detection in US Videos (Lin et al., 2022) | Breast benign Brest malignant | Breast lesion classification | Diag | 171 | - |
| Breast US Images Dataset (Al-Dhabyani et al., 2020) | Breast | Breast cancer level classification Breast tumour localisation Brest Diag view identification | Diag VRA LL | 210 | CC0: PD |

(Continued) Table 7

| Dataset | Anatomy | Clinical scenarios | Task | Case | License |
|---|---|---|---|---|---|
| Dermatologic Ultrasound Images for classification (Laverde Saad et al., 2021) | Skin | Skin tumor level classification | Diag | 100 | - |
| Polycystic Ovary Ultrasound Images Dataset (Wisesty et al., 2018) | Ovary | Polycystic Ovary Syndrome localisation | VRA | 10 | CC0: PDD |
| CUBS (Meiburger et al., 2021) | Carotid | Carotid thickness estimation
Carotid detection
Catotid Diag view identification | VRA
OD
CVE | 681 | CCA 4.0I |
| Knee US dataset in a population-based cohort (Novin et al., 2023) | Knee | Knee US KL and pain grad classification
Knee Diag view identification
Knee lesion localisation | Diag
VRA
OD | 326 | CC0 1.0 |
| HC18 (van den Heuvel et al., 2018) | Fetal head | Fetal head circumference estimation
Fetal head detection | OD
CVE | 202 | CCA 4.0I |
| KFGNet (NeuronXJTU & palkia1998, 2023) | Thyroid | Thyroid nodule level classification
Thyroid nodule localisation | Diag
LL | 206 | - |
| Thyroid (Krönke et al., 2022) | Thyroid Left
Thyroid right | Thyroid Diag view identification | VRA | 563 | CC BY |
| GDPHSYSUCC (Mo█ et al., 2022) | Breast | Breast lesion classification | Diag | 109 | - |
| LEPset (Li et al., 2023b) | Pancreas | Pancreatic cancer classification | Diag | 101 | CCA 4.0I |
| COVID-BLUES (Wiedemann et al., 2025) | Lung | COVID-19 level classification
Lung US caption generation
Lung Diag view identification | Diag
VRA
CG | 318 | ANN 4.0 I |
| Ultrasound Guided Regional Anesthesia (Tyagi et al., 2024) | Brachial plexus | Brachial plexus detection | OD | 179 | Non-commerical |
| Unity Imaging Collaborative (Shun-Shin, 2023) | Cardiac | Caridac Keypoint Detection | KD | 500 | CCANN 4.0 I |

(Continued) Table 7

| Dataset | Anatomy | Clinical scenarios | Task | Case | License |
|---------|---------|-------------------|------|------|---------|
| C-TRUS Dataset (Leenings et al., 2025) | Colon | Colon wall detection | OD | 166 | - |
| ACOUSLIC-AI (Sappia, 2024) | Fetal abdominal | Fetal abdominal circumference estimation
Fetal adominal OD | VRA
OD
CVE | 310 | CCANCSA 4.0I |
| PSFHS (Bai, 2024) | Fetal head
Fetal pubic symphysis | Fetal head detection
Fetal pubic symphysis detection | OD | 100 | CCA 4.0I |
| JNU-IFM (Lu et al., 2022) | Fetal head
Fetal pubic symphysis | Fetal view identification
Fetal head detection
Fetal pubic symphysis detection | VRA
OD | 202 | CC BY 4.0 |
| Dataset of B-mode fatty liver US images (Byra et al., 2018) | Liver | Liver steatosis classification
Liver fat value estimation
Liver Diag view identification | Diag
VRA
CVE | 222 | CCA 4.0I |
| African Fetal Standard Plane (Sendra-Balcells et al., 2023) | Fetal abdomen
Fetal brain
Fetal femur
Fetal thorax | Fetal standard plane identification | VRA | 10 | CCA 4.0I |
| BrEaST (Pawłowska et al., 2024) | Breast | Breast LL | LL | 100 | CC BY 4.0 |
| Ultrasound Breast Images for Breast Cancer (Sairam, 2020) | Breast | Breast cancer classification | Diag | 100 | CC0: PD |
| US simulation and segmentation (Vitale et al., 2020) | Abdominal | Abdominal OD | OD | 100 | - |
| Carotid Artery Ultrasound and Color Doppler (Pahuni Choudhary, 2023) | External carotid
left carotid
right carotid | Carotid Diag view identification | VRA | 100 | Apache 2.0 |
| AUITD (Maroua, 2020) | Thyroid | Thyroid lesion classification | Diag | 100 | - |
| Auto-PCOS classification (Maroua, 2020) | Ovary | Polycystic Ovary Syndrome classification
Ploycystic Diag view identification | Diag
VRA | 218 | CCA 4.0I |
| Auto-PCOS classification (Maroua, 2020) | Ovary | Polycystic Ovary Syndrome classification | Diag | 100 | CC BY 4.0 |

**Summary of Dataset Licensing Terms.** The datasets included in **U2-BENCH** span a range of open and restricted licenses. For clarity, we summarise the licensing terms here.

- **CC0 / Public Domain (PD, PDD):** Fully open; free use, modification, and redistribution without attribution, including commercial use.

- **CC BY / CC BY 4.0 / CCA 4.0I:** Free use with attribution; permits modification and redistribution.

- **CC BY-NC-SA / CC BY-NC-SA 4.0 / variants written as ANN 4.0 I, CCANN 4.0 I, CCANCSA 4.0I:** Non-commercial use only; derivatives must adopt the same license.

- **CC BY-NC-ND:** Attribution required; non-commercial; no derivatives permitted. Minor naming variations follow the dataset providers' release notes.

- **MIT License:** Permissive license allowing free use, modification, and redistribution, including commercial applications.

- **Apache 2.0:** Permissive license with an explicit patent grant.

- **Non-commercial data use agreement:** Access provided strictly for non-commercial research; redistribution or reuse requires separate permission.

- **Unspecified / "–":** Publicly released datasets without an explicit license. Usage follows the terms communicated by the original authors.

