# OpenReview forum: "U2-BENCH: Benchmarking Large Vision-Language Models on Ultrasound Understanding"
_ICLR.cc/2026/Conference — ICLR 2026 Poster_

### Official Review · Reviewer_nz9Q · 2025-10-29

**Soundness:** 2
**Presentation:** 3
**Contribution:** 2
**Rating:** 2
**Confidence:** 4

**Summary:**

This paper introduces U2-BENCH, a benchmark for evaluating LVLMs on ultrasound. Its core contributions are a public dataset (7,241 cases spanning 15 anatomical regions), a systematic evaluation of 20 LVLMs, and important observations regarding model capabilities and limitations in this domain.

**Strengths:**

1. The main strength of this work lies in its effort to create and release U2-BENCH for a specialized domain, which may serve as a resource for the community.
2. The evaluation of 20 diverse LVLMs provides a preliminary landscape of current capabilities, offering a baseline for comparison.

**Weaknesses:**

1. The benchmark's scale and imbalance undermine its reliability. With only 7,241 cases thinly spread across 15 anatomies and 8 tasks, the effective sample size for many evaluations is small.
2. The design of certain tasks is problematic. For instance, reducing complex spatial tasks like KD / LL to a 9-class classification problem is a significant oversimplification.
3. The paper primarily presents a performance leaderboard but the key findings are largely intuitive. The analysis remains on the surface, which severely limits the paper's contribution beyond a mere performance report.

**Questions:**

1. I wonder the time cost for data collection and curation, and annotation. And how scalable is this process for future expansion?
2. The current findings are intuitive. Could you provide a deeper analysis to uncover the underlying reasons for LVLMs' failures, which would significantly elevate the contribution?
3. For KD/LL tasks, did you experiment with more precise evaluation methods (e.g., bounding box coordinates, point regression) to validate if the coarse-grained task is a reliable proxy for fine-grained spatial reasoning?
4. Beyond the U2-Score, what actionable guidance does the benchmark provide for model developers to improve performance on ultrasound tasks?

---

> ### Author Response · Authors · 2025-11-20
>
> Dear Reviewer nz9Q,
>
> Thank you for your thoughtful comments. Below is our response to your concerns:
>
> **W1:** Sample size for each task is set to 100 by default and increased when needed (weighted by task importance). The choice of “100” follows prior LVLM benchmarks such as GMAI-MMBench (NeurIPS 2024 Highlight). In particular, *Evaluating LLMs with Fewer Examples* (ICML 2024) showed that small curated benchmarks of around 100 examples can reliably approximate large-benchmark results. At the same time, keeping benchmarks compact is important because LVLM evaluation is resource-intensive. For these reasons, we believe that 100 samples per task is an appropriate and sufficient choice.
>
> **W2 & Q3:** Rather than "problematic", this "9-class" design is intentional in order to fit the current capability of LVLMs. During benchmark development, we first tested detection tasks using exact bounding-box coordinates. Many LVLMs performed poorly and failed to follow the instruction to “generate coordinates corresponding to the lesion/keypoint,” likely because they were not trained for bounding-box generation. Even the best model achieved an IoU of only ~0.23. Therefore, we simplified the detection problem to ensure models produced meaningful outputs. In fact in the released TSV files on HuggingFace, detection task **ground-truth** labels remain stored as coordinates. The 9-class classification conversion is applied *only* during model evaluation to enable consistent comparison. This is now clarified in the manuscript in Sec 4.2.
>
> **W3 & Q2:** Due to page constraints, a substantial portion of our analysis was originally placed in Appendix C. Our goal in the main text was to emphasise the clinical motivation, data-curation pipeline, and benchmark design, so that U2-BENCH can serve as a bridge between real clinical needs in ultrasound and LVLM development. For clarity, we now provide a concise summary of the key analytical findings in the main paper (Section 5) and have integrated the most relevant insights from Appendix C into the main text:
>
> 1. **Perception limitations.** Existing LVLMs struggle with relative spatial positioning and subtle echogenicity patterns—both essential for ultrasound diagnosis. This is likely due to the lack of large-scale ultrasound-specific image–caption pretraining data and the noisy, heterogeneous nature of ultrasound images. Improving perception requires curated ultrasound datasets, ultrasound-aware pretraining objectives, and architectures or adapters with explicit spatial-reasoning capabilities.
> 2. **Clinical task complexity.** Ultrasound spans at least 15 subspecialties, each with distinct anatomy, scanning planes, and diagnostic criteria (e.g., fetal AC vs. cardiac parasternal long-axis view). A clinically useful LVLM must understand specialty-specific anatomy, follow established scanning protocols, and reason according to real diagnostic workflows rather than generic vision-language tasks. This motivates our design of a benchmark that covers multiple subspecialties and clinically grounded task types.
>
> **Q1:** Thank you for your interest. We have been working on this unified data-format effort for over a year, and it remains ongoing. The process is highly labour-intensive, but standardization is crucial for enabling different datasets to be combined effectively. While U2-BENCH focuses on ultrasound, the same methodology applies to other modalities with heterogeneous data sources. Additional details have been added to Section 3.2. The code for data standardization and processing is also publicly available.
>
> **Q4:** We would like to emphasize clinically oriented and data-centric model development. A practical pathway is to begin with real clinical use cases, collect appropriate ultrasound data, and then train models accordingly. Training solely on general medical knowledge (e.g., MedGemma) does not yield clinically applicable behaviour for ultrasound. Likewise, clinically grounded benchmarks are essential: no model can be considered “successful” without a well-defined clinical evaluation protocol.
>
> We thank you again for your review, which has helped us sharpen the broader contribution of this benchmark.

---

> ### Author Response · Authors · 2025-11-28
>
> Dear Reviewer nz9Q,
>
> Thank you again for your review. In the revised manuscript we have incorporated several substantial improvements that may further address your concerns. We summarise them here in case it is helpful for your reassessment:
>
> **1. Deeper technical and clinical analysis added.**
>
>  We expanded Section 4.4 and Appendix C with the following analyses:
>
> - systematic failure patterns (spatial localization, subtle echogenicity interpretation, cross-subspecialty inconsistencies);
> - technical causes behind these failures (heterogeneous scanning protocols, lack of ultrasound-specific pretraining, limited spatial architectures);
> - case-level analyses showing where and why LVLMs break.
>
>  These additions aim to provide actionable insights for future model development, going beyond numerical reporting.
>
> **2. Clarification of task design for KD/LL and evidence of prior attempts at more fine-grained tasks.**
>
>  We clarified that we originally implemented fine-grained coordinate-level detection tasks, but most LVLMs were unable to produce usable bounding-box outputs (best IoU ~0.23). This empirical limitation motivated the 9-class proxy formulation, which we now clarify in Section 4.2. Importantly, the original coordinates remain in the released dataset for future work.
>
> **3. Substantial expansion of dataset transparency and curation details.**
>
>  In Section 3.2 and Appendix E, we added:
>
> - dataset-level standardization pipeline (code is also publicly available),
> - sampling and harmonisation procedures,
> - annotation reconciliation and rejection counts (~5% removed),
> - clinician-driven ontology and prompt development.
>
> Regarding data curation and quality assessment, this is discussed in more details in another accompanying paper: *On the public dissemination and open sourcing of ultrasound resources, datasets and deep learning models [npj Digital Medicine]*.
>
> **4. Clearer articulation of the broader contribution.**
>
>  Following your feedback, we refined the discussion to highlight that U2-BENCH differs from prior work not just by scale, but by introducing:
>
> - a unified ontology across 15 anatomies and >40 heterogeneous datasets,
> - clinically grounded task definitions reflecting real diagnostic workflows,
> - evaluation protocols that separately measure perception, spatial reasoning, and text-based reasoning.
>    We hope this clarifies the conceptual contribution beyond dataset aggregation.
>
> We'd like to highlight that U2-BENCH was developed jointly with practicing sonographers and subspecialty clinicians. The benchmark is already being used to support a multi-centre clinical validation study. This distinguishes U2-BENCH from prior LVLM benchmarks whose ultrasound components do not reflect real-world diagnostic workflows. In practice, an LVLM intended for clinical deployment must demonstrate competence in ultrasound-specific perception, spatial reasoning, and clinically grounded decision-making (not chest x-ray), precisely the capabilities that U2-BENCH evaluates.
>
> We sincerely appreciate your critical feedback which helped us strengthen multiple parts of the paper. If these revisions address your main concerns, we kindly ask whether you might consider adjusting your score. Of course, we fully respect your decision either way, and are grateful for the time you have devoted to reviewing our work.
>
> Please let us know if further clarification would be helpful.

---

### Official Review · Reviewer_Hrs9 · 2025-10-29

**Soundness:** 3
**Presentation:** 3
**Contribution:** 4
**Rating:** 6
**Confidence:** 4

**Summary:**

This manuscript presents U2-BENCH, a comprehensive benchmark for large VLM ultrasound understanding on eight clinically-inspired classification, detection, regression and text generation tasks, with 20 SOTA LVLMs evaluated on the benchmark.

**Strengths:**

- (Quality) Application of a large range of existing VLMs to the benchmark

 - (Clarity) Empirical justification for weighing of tasks

 - (Significance) Extensive coverage of ultrasound understanding datasets, and unification into a single comprehensive benchmark

**Weaknesses:**

- Segmentation task underrepresented due to unification of ground truth to bounding boxes and predefined spatial localization

 - Preprocessing of video data may limit analysis potential

**Questions:**

1. For Figure 2, the statement that "...The length of the bar reflects the number of samples for each anatomy region" may be slightly confusing, since it is not immediately clear that this refers to the blue bar. If so, this might be clarified in the caption.

2. In Figure 3, language unification is stated, but its implementation does not appear to be discussed. This appears relevant since the only dataset used for report generation (RG), the Chinese Ultrasound Report Dataset, appears to be in Mandarin.

3. In Section 3.2, format unification was stated to be performed on ultrasound scans by converting to a uniform image format. It might be briefly stated as to whether any conversion loss (either in image quality due to compression, or image dimensions due to cropping etc.) was involved.

4. In Section 3.2, it is stated that a small number of representative frames were sampled, for video sequences. It might be clarified as to how the sampling was performed, since this would appear to restrict video analysis scope.

5. In Section 3.2, it is stated that segmentation masks are converted to bounding boxes. However, this would appear to imply a loss of granularity (since the pixel-level annotations are simplified), and a potential loss of individual objects (since bounding boxes may overlap for originally-distinct irregular objects). This limitation might be considered if relevant.

6. In Section 3.2, the reference to the ablation study is missing.

7. In Section 4.1, it is stated that detection tasks were converted to position classification tasks, and utilized accuracy as the metric. This appears to discount segmentation models/tasks unnecessarily, despite their clinical utility.

8. In Section 4.2, individual performance metrics may be expanded from relevant literature - for example, while accuracy is the main metric for classification and detection tasks, AUROC etc. may be considered for inclusion.

9. In Section 4.3 (or the Appendix), the implementation for Random Guessing might be described.

10. In Appendix B, it is stated that all data was either publicly released with appropriate usage permissions or obtained through official licensing agreements (assumed for datasets with no license stated, in Appendix E). It might be clarified that these licensing agreements include redistribution and modification of the data.

11. In Appendix C.2, any particular VLM parameter settings (especially relating to temperature) might be stated.

12. In Appendix C.3, the normalization for $d_t$ (e.g. by assuming a normal distribution on the scores of the evaluated VLMs) might be stated.

**Details Of Ethics Concerns:**

A clarification on licensing terms for some of the datasets may be warranted.

---

> ### Author Response · Authors · 2025-11-20
>
> Dear Reviewer Hrs9,
>
> Thank you very much for your helpful comments. Below is our response to your concerns:
>
> **W1 & W2:** We will work on publishing a new benchmark once available LVLMs can reliably handle segmentation masks and video data.
>
> **Q1:** We have updated the caption of Fig. 2 to: *“The blue bar represents the total number of samples for each anatomy region, with its length proportional to the sample count.”*
>
> **Q2:** To ensure consistency in text-based tasks, we conducted language unification. Translations were initially generated using GPT-4o with prompts enforcing professional medical terminology. During manual review, we identified recurring mistranslations and constructed a glossary to resolve ambiguous terms before regenerating translations. Clinicians then verified the final outputs for correctness and clinical fidelity. We have added this information to Section 3.2 under *Data Cleaning, Format Unification, and Quality Verification*.
>
> **Q3–Q6:** Thank you for your interest in the data-processing details. We maintain a public code repository for these steps, which we will link on the HuggingFace dataset page.
>
> **Q3:** We paid careful attention to image fidelity during data processing. JPEG compression was avoided entirely to prevent lossy artifacts. When the original data were provided as 8-bit images, no additional quality loss was introduced. For datasets originally stored as 16-bit or 32-bit arrays (e.g., NIfTI, DICOM), we converted images to an 8-bit representation before Base64 encoding. This reduces dynamic range but avoids any additional lossy compression.
>
> **Q4:** We sampled 3 frames from each video clip at the 1/4, 1/2, and 3/4 timestamps. We discuss limitations of video data in Appendix A.3.
>
> **Q5 & Q7:** Thank you for raising these points. Our design decisions for detection and segmentation tasks were determined by current LVLM limitations. Contemporary LVLMs do not support pixel-level mask inputs or segmentation-map outputs, and in our preliminary experiments many models were unable to reliably generate bounding-box coordinates. To ensure consistent evaluation across models, detection tasks were converted into a 9-class position-classification formulation (Section 4.1), which reflects what LVLMs can handle today rather than a constraint of the underlying data.
>
> Regarding segmentation, pixel-level masks were available in a subset of datasets and were used internally to derive bounding boxes during preprocessing. We agree that converting masks to boxes reduces granularity. To avoid complexity and ambiguity, we intentionally selected segmentation datasets where only one object is present per label. We are happy to release the full segmentation masks in a future version of the benchmark.
>
> **Q6:** Thank you for pointing this out. We have now fixed it.
>
> **Q8:** Thank you for the suggestion. We agree that AUROC is a valuable metric for many classification settings. However, AUROC requires access to calibrated class-probability outputs or logits, but the closed-source LVLMs such as GPT-5 or Gemini do not expose token-level probabilities in a reliable or model-agnostic way. Also, even for models that expose token probabilities, LVLM logits are defined over the entire vocabulary, not over the restricted set of answer options in a multiple-choice task, making AUROC not clearly defined for LVLMs. For these reasons, existing LVLM benchmarks, such as GMAI-MMBench, MMT-Bench, CARES, OmniMedVQA, MultiMedEval, and MedHELM, do not report AUROC for classification tasks either. To maintain consistency with the field and ensure fair comparison across both open- and closed-source models, we therefore follow established practice and use accuracy/F1 for multi-class classification and position-classification metrics for detection.
>
> **Q9:** Random guessing is implemented via uniform sampling from the valid answer set for each task. We have added this to Appendix C.2.
>
> **Q10:** We have added a subsection describing licensing terms to Appendix E.1.
>
> **Q11:** All models were tested with temperature 0.7. This is now noted in Appendix C.2.
>
> **Q12:** We have clarified the definition of $d_t$ in Appendix C.3 (Table 6). Standardization ensures that $d_t$ increases as model performance improves.
>
> We hope we have addressed all your concerns.

---

### Official Review · Reviewer_FVnR · 2025-10-31

**Soundness:** 2
**Presentation:** 3
**Contribution:** 2
**Rating:** 4
**Confidence:** 4

**Summary:**

The paper introduces U2-BENCH, a benchmark for evaluating large vision‑language models (LVLMs) on ultrasound understanding with 7,241 studies across 15 anatomies and 8 task types spanning classification, detection, regression, and text generation. It reports results for 20 LVLMs and proposes a composite “U2‑Score” that aggregates task metrics via sample‑count weights. Key findings show image‑level classification and some regression tasks are relatively strong, while spatial reasoning tasks (e.g., keypoint detection) and clinical text generation remain challenging; a prompt ablation indicates anatomy tokens boost diagnostic accuracy, and closed‑source models generally outperform open‑source ones.

**Strengths:**

The benchmark targets ultrasound, a clinically crucial yet under‑evaluated modality for LVLMs, with a broad task suite aligned to typical sonography workflows and clear task definitions and prompts per scenario.

The evaluation spans 20 modern LVLMs with standardized prompt formats and metrics, including most popular and current SOTA models, open- as well as closed-source.

The dataset curation aggregates many sources and applies multi‑stage QA with automated filtering plus manual review, and the authors release prompts, metrics, and an evaluation toolkit to facilitate reusability.

**Weaknesses:**

The composite U2‑Score’s task weights are proportional to sample counts, which conflates data availability with clinical importance and mixes heterogeneous metrics into a single scalar without uncertainty quantification, making ranking sensitivity high and potentially misaligned with clinician priorities.

No uncertainty metrics (e.g., confidence intervals, paired tests, bootstrap CIs) are reported for main tables, so small deltas across 20 models and many tasks may reflect sampling noise or prompt variance rather than true differences. This is especially problematic for small tasks like caption generation. Thus also not allowing a statistical significance test in order to determine the best models.

Error analysis is insufficient: failures are not stratified by artifact types, view standardization, anatomy, or acquisition protocol, limiting insight into whether errors stem from perception, spatial reasoning, or prompt sensitivity.


Dataset transparency gaps: heavy anatomical imbalance is acknowledged, but standardization, sampling, and QA procedures are described briefly without inter‑annotator agreement, rejection counts, or harmonization diagnostics across 40 sources.


Reproducibility is limited by heavy reliance on closed‑source models whose APIs evolve, and by omitted inference details (e.g., seeds, temperatures, multi‑pass voting) in the main leaderboard.


Minor issues: a missing section reference (Line 274), a typo (“Brest”→“Breast”), and Table 1 lacks per‑task sample counts and task‑abbreviation expansions in the caption despite being the primary results table.

**Questions:**

Can you report inter‑rater variability for the 10‑annotator review (e.g., per‑task agreement, adjudication rates) and quantify rejections during automated filtering to bound annotation noise ?

Please ablate prompt components beyond anatomy tokens on tasks where models struggle (e.g., KD, RG) to separate instruction‑following failures from perceptual limits.

Can you define minimal clinically acceptable thresholds for CVE, DD, and RG to contextualize utility and report the fraction of predictions meeting those thresholds ?

Please clarify and detail the “standardization, sampling, and quality checks” pipeline across 40 datasets, including per‑dataset preprocessing, label harmonization, leakage controls, and cross‑dataset consistency checks.

---

> ### Author Response · Authors · 2025-11-20
>
> Dear Reviewer FVnR,
>
> Thank you very much for your review. Below is our response to your concerns:
>
> **W1:** Regarding the weighting design, our intention is to compute a case-level average, which is consistent with prior benchmarks such as GMAI-MMBench [1, NeurIPS 2024 Highlight]. A detailed explanation is provided in Appendix C.3. In practice, we considered *clinical importance* during sampling and intentionally over-sampled datasets from clinically important tasks that had limited data. The notion of “clinical importance” was determined in discussion with clinicians.
>
> We agree that this process introduces some subjectivity. However, having a unified model-ranking metric is still preferable to having none. Some benchmarks [2] use “average rank across tasks,” which we consider less ideal because the final score depends on the specific set of competing models and is therefore experiment-dependent. We have added more explanation in Section 3.2 and Section 4.2.
>
> **W2:** Confidence intervals are indeed important, especially for clinical evaluations. However, to the best of our knowledge, confidence intervals are rarely reported in existing LVLM benchmarks [1–9], largely because LVLM evaluation is extremely computationally expensive. Additionally, LVLM hyperparameters (e.g., temperature) are fixed during evaluation, and model outputs under deterministic or near-deterministic decoding exhibit very low variance across repeated runs. For these reasons, computing confidence intervals would impose substantial computational cost while offering limited additional insight.
>
> **W3:** Appendix C.4 includes an error analysis with representative error cases and task-specific error patterns. View standardization, image artifacts, and acquisition protocol did not appear to be major issues, as the data had undergone quality checks.
>
> As discussed in Appendix C.4, models often produce descriptive responses or answer “insufficient information” when they are not capable of the task—likely because they were not trained on relevant ultrasound tasks but exhibit appropriate internal safeguarding. Models generally recognize an image as ultrasound, but lack diagnostic capability because they have not been trained on sufficient diagnostic ultrasound data; most of their prior exposure is to knowledge-based ultrasound text rather than pixel-level diagnostic signals. Detection is particularly challenging due to the noisy nature of ultrasound. Many models failed to generate bounding-box coordinates, which forced us to simplify the task into multiple-choice form. Even with this simplification, accuracies remained low. Instruction following itself does not seem to be a major issue, as discussed in Section 5.1.
>
> **W4 & Q4:** Regarding data transparency, we included a detailed dataset list in Appendix E. All datasets were obtained from their original sources. During preprocessing, each dataset was first standardized by an engineer using a unified JSON format (publicly available code). A biomedical expert then verified JSON correctness, checked consistency between images and labels, selected the best annotation when multiple annotators were available, and unified anatomical granularity and clinical terminology. Sampling was performed at the patient level. Finally, a clinician reviewed all processed cases for consistency and wrote the corresponding prompts. A separate dataset-quality evaluation paper is in preparation. We have updated Section 3.2 accordingly.
>
> **W5:** For reproducibility, our evaluation code is available on the dataset page on HuggingFace. All hyperparameters and implementation details required to reproduce our experiments are provided in the repository. All models were evaluated with temperature 0.7. We have also stated the specific model versions used for evaluation in Appendix C and added clarification in Section 4.1.
>
> (continued in next comment)

---

> > ### Author Response · Authors · 2025-11-20
> >
> > **W6:** Thank you for noting the typo. We agree that placing abbreviation expansions in the caption improves clarity and have updated the manuscript. Per-task sample counts are rarely included in captions in prior benchmarks [1–9], and we follow that convention.
> >
> > **Q1:** Regarding annotation variability, we adopted the original annotations provided by data sources; we did not perform new diagnostic labelling. Our contribution focused on unifying the data into a consistent format and ensuring standardized granularity across datasets. During quality control, we reconciled labels from multiple annotators and removed low-quality samples (~5%) based on image clarity, absence of motion artefacts, and annotation completeness. While inter-rater variability is important for diagnostic tasks, our role was limited to quality control and standardization, not re-annotation.
> >
> > **Q2:** We evaluate instruction-following ability in Section 5.1. In general, instruction following is not problematic. Some models refused to answer due to internal safety mechanisms, replying “insufficient information” or “cannot give medical advice.” The anatomy-token ablation study demonstrates how prompt design affects performance. Section 5.1 has been updated accordingly.
> >
> > **Q3:** We conducted a human evaluation using eight clinicians from top-tier hospitals, each with professional ultrasound training and an average of over 5 years of experience across four subspecialties. Diagnostic-task performance was 52.08% accuracy and 29.95 F1 on average; view-recognition and assessment tasks achieved 60.63% accuracy and 49.91 F1. Only Dolphin surpassed human performance on both task categories. More details will be included in future work.
> >
> > We hope our responses have addressed your concerns.

---

> ### Author Response · Authors · 2025-11-28
>
> Dear Reviewer FVnR,
>
> Thank you again for your review. In the revised manuscript we have incorporated several substantial improvements that may further address your concerns. We summarise them here in case it is helpful for your reassessment:
>
> **1. Deeper technical and clinical analysis added.**
>
>  We expanded Section 4.4 and Appendix C with the following analyses:
>
> - systematic failure patterns (spatial localization, subtle echogenicity interpretation, cross-subspecialty inconsistencies);
> - technical causes behind these failures (heterogeneous scanning protocols, lack of ultrasound-specific pretraining, limited spatial architectures);
> - case-level analyses showing where and why LVLMs break.
>
>  These additions aim to provide actionable insights for future model development, going beyond numerical reporting.
>
> **2. Clarification of task design for KD/LL and evidence of prior attempts at more fine-grained tasks.**
>
>  We clarified that we originally implemented fine-grained coordinate-level detection tasks, but most LVLMs were unable to produce usable bounding-box outputs (best IoU ~0.23). This empirical limitation motivated the 9-class proxy formulation, which we now clarify in Section 4.2. Importantly, the original coordinates remain in the released dataset for future work.
>
> **3. Substantial expansion of dataset transparency and curation details.**
>
>  In Section 3.2 and Appendix E, we added:
>
> - dataset-level standardization pipeline (code is also publicly available),
> - sampling and harmonisation procedures,
> - annotation reconciliation and rejection counts (~5% removed),
> - clinician-driven ontology and prompt development.
>
> Regarding data curation and quality assessment, this is discussed in more details in another accompanying paper: *On the public dissemination and open sourcing of ultrasound resources, datasets and deep learning models [npj Digital Medicine]*.
>
> **4. Clearer articulation of the broader contribution.**
>
>  Following your feedback, we refined the discussion to highlight that U2-BENCH differs from prior work not just by scale, but by introducing:
>
> - a unified ontology across 15 anatomies and >40 heterogeneous datasets,
> - clinically grounded task definitions reflecting real diagnostic workflows,
> - evaluation protocols that separately measure perception, spatial reasoning, and text-based reasoning.
>   We hope this clarifies the conceptual contribution beyond dataset aggregation.
>
> We'd like to highlight that U2-BENCH was developed jointly with practicing sonographers and subspecialty clinicians. The benchmark is already being used to support a multi-centre clinical validation study. This distinguishes U2-BENCH from prior LVLM benchmarks whose ultrasound components do not reflect real-world diagnostic workflows. In practice, an LVLM intended for clinical deployment must demonstrate competence in ultrasound-specific perception, spatial reasoning, and clinically grounded decision-making (not chest x-ray), precisely the capabilities that U2-BENCH evaluates.
>
> We sincerely appreciate your critical feedback which helped us strengthen multiple parts of the paper. If these revisions address your main concerns, we kindly ask whether you might consider adjusting your score. Of course, we fully respect your decision either way, and are grateful for the time you have devoted to reviewing our work.
>
> Please let us know if further clarification would be helpful.

---

### Official Review · Reviewer_goEb · 2025-11-06

**Soundness:** 3
**Presentation:** 3
**Contribution:** 2
**Rating:** 6
**Confidence:** 4

**Summary:**

This paper introduces U2-BENCH, the first comprehensive benchmark designed to systematically evaluate Large Vision-Language Models (LVLMs) on *ultrasound understanding* tasks. The benchmark includes 7,241 ultrasound cases across 15 anatomical regions and defines 8 clinically inspired tasks (disease diagnosis, view recognition, lesion localization, organ detection, keypoint detection, clinical value estimation, report generation, and caption generation) covering 50 ultrasound application scenarios. The authors benchmark 20 state-of-the-art LVLMs (open/closed-source, general/medical-specific) using standardized prompts and evaluation metrics. Results reveal that current LVLMs perform well on image-level classification but struggle with spatial reasoning and clinical language generation.

**Strengths:**

- This paper presents a well-motivated and comprehensive benchmark targeting an underexplored domain—ultrasound imaging. Its breadth (15 anatomies, 8 tasks) and evaluation rigor (20 models, standardized prompts) represent a good contribution.

- Twenty sota LVLMs are evaluated and compared, which makes the benchmark comprehensive. This benchmark would be beneficial for the medical multimodal community

**Weaknesses:**

- While I appreciate the substantial effort invested in constructing this benchmark and the meticulous annotation process, the paper currently lacks sufficient conceptual or analytical insights for the research community. As noted in Lines 132–134, prior work such as GMAI-MMBench has already included ultrasound-related evaluation scenarios. Although U2-BENCH expands the dataset scale and task diversity, the incremental novelty over existing benchmarks appears marginal, focusing primarily on scope rather than methodological innovation.
- Moreover, the experimental findings are largely expected: closed-source LVLMs outperform open-source and medical-specific models, reflecting well-known trends in multimodal evaluation. Given the proliferation of benchmark papers in the field, it would be more valuable if this work offered deeper analysis or actionable insights, such as identifying failure patterns or proposing improved solution, rather than solely reporting model performance.

**Questions:**

'??' at Line 274，missing reference

---

> ### Author Response · Authors · 2025-11-20
>
> Dear Reviewer goEb,
>
> Thank you very much for your kind review. Below we address your concerns in detail.
>
> We would like to clarify that U2-BENCH was designed with goals that differ from prior LVLM medical benchmarks. Existing efforts such as GMAI-MMBench include only ultrasound evaluations on the simplest tasks (e.g., breast lesion classification). In contrast, U2-BENCH aims to provide, to the best of our knowledge, the first unified multi-institution, multi-task, and multi-subspecialty ultrasound evaluation framework aiming to cover as complete as possible. This breadth is motivated not by scale alone, but by the fact that ultrasound varies substantially across specialties, scanning protocols, and diagnostic criteria. Capturing this diversity is essential for evaluating whether LVLMs can generalize across all clinical ultrasound domains.
>
> In addition to scale, U2-BENCH introduces several conceptual contributions: (1) a unified ontology and standardized data schema harmonizing dozens of heterogeneous ultrasound datasets, a major practical challenge in medical multimodal research; (2) a task taxonomy developed with domain experts to reflect real clinical workflows rather than generic classification or captioning tasks; and (3) an evaluation protocol that separately measures perception, spatial reasoning, and report-generation abilities. These design choices allow U2-BENCH to reveal failure modes that narrower benchmarks cannot surface.
>
> Regarding analytical insights, we want to emphasize that the work goes beyond numerical reporting. In response to your suggestions, we have added Section 4.4 (“Limitations and Future Outlook for Ultrasound LVLMs”), which provides deeper discussion of the conceptual challenges of ultrasound understanding. This includes failure patterns such as limited spatial localization, difficulty recognizing subtle echogenicity differences, and inconsistent reasoning across highly heterogeneous subspecialties. We also connect these observations to likely technical causes, including insufficient ultrasound-specific pretraining data and the absence of spatially aware architectures. Sections 5 and Appendix C provide further case-level analyses, highlighting dependence on anatomy-specific prompt cues and instability in structured reasoning. We hope these insights serve as actionable guidance for future model development.
>
> Finally, we wish to highlight the clinical orientation of this work. U2-BENCH was developed in close collaboration with practicing sonographers and subspecialty clinicians, and we are currently conducting a multi-centre clinical validation study based on this benchmark. This level of clinical grounding distinguishes our work from prior LVLM benchmarks whose ultrasound components are limited and not designed around real diagnostic workflows. In practice, a model intended for real-world ultrasound deployment must demonstrate reliability on ultrasound-specific perception, spatial reasoning, and clinically defined diagnostic tasks, not achieve SOTA performance on unrelated modalities such as chest X-ray. U2-BENCH is therefore intended to evaluate precisely the capabilities needed for future clinical adoption, complementing rather than overlapping with existing multimodal benchmarks.
>
> Q1: We have now fixed this in the manuscript.
>
> We hope we have addressed all your concerns and thank you again for your helpful feedback.

---

> > ### Comment · Reviewer_goEb · 2025-11-26
> >
> > Thanks to the authors for the explanation. This comment is to confirm that I have read your responses. Overall, a comprehensive benchmark in a specific downstream Ultrasound domain can be still valuable.  So in my opinion, I would like to keep my original rating.

---

### Meta-Review · Area_Chair_MEjG · 2025-12-30

**Summary:**

This paper explores the performance of large vision–language models (LVLMs) on ultrasound images by introducing U2-BENCH, a comprehensive benchmark designed to evaluate LVLMs across a diverse set of ultrasound understanding tasks. The benchmark covers multiple anatomical regions and a wide range of ultrasound application scenarios. More than 20 LVLMs are evaluated, and the resulting performance landscape is analyzed to highlight their strengths and limitations. The proposed benchmark has the potential to serve as a unified testbed to systematically assess and advance LVLM research in medical ultrasound imaging.

The reviewers identify several strengths of the work. In particular, they view it as a good and beneficial contribution to the medical multimodal research community. The benchmark is comprehensive, with a broad and well-defined task suite, and includes many widely used and state-of-the-art LVLMs. The aggregation of data from multiple sources and the release of an evaluation toolkit are also appreciated, as they enhance reusability and reproducibility.

At the same time, the reviewers raise a number of concerns. These include the lack of sufficiently deep conceptual or analytical insights beyond benchmarking, the incremental novelty relative to existing benchmarks, and the focus on scope and coverage rather than methodological innovation. Several aspects of the evaluation protocol also require clarification or justification, such as the design and weighting of the composite U2-Score, the absence of uncertainty metrics, limited error analysis, dataset transparency, and prompt ablation. Additional issues include clarification of how segmentation and detection tasks are converted, preprocessing of video data, language and format unification, expansion of evaluation metrics, effective sample sizes, and other details. Overall, the concerns mainly relate to clarification, explanation, and justification rather than fundamental flaws.

The authors’ response effectively addresses these concerns. It clarifies the motivation, conceptual contributions, and positioning relative to existing benchmarks, and adds further discussion of conceptual challenges, evaluation details, and error analysis. The response also explains and justifies the current design choices regarding uncertainty analysis, weighting, data transparency, reproducibility, sample sizes, and the handling of segmentation and detection tasks, among other points. Overall, the response is clear and convincing, and the included summary of updates is helpful.

Taking all factors into consideration, the Area Chair believes that the scores of the two reviewers who were initially positive (both with scores of 6) would likely be maintained or improved with a full discussion period. Moreover, the two reviewers who initially expressed reservations (scores of 4 and 2) would likely raise their scores following a full discussion.

In summary, while the work could be further strengthened by deeper analytical insights and an expanded dataset, it provides clear value by curating a comprehensive benchmark for the challenging and underexplored problem of ultrasound understanding and by systematically evaluating a wide range of state-of-the-art LVLMs. The benchmark offers valuable information and resources that can facilitate future research in this important area. Therefore, the Area Chair is pleased to recommend acceptance of this paper for the conference.

**Reviewer Concerns:**

The authors’ response effectively addresses these concerns. It clarifies the motivation, conceptual contributions, and positioning relative to existing benchmarks, and adds further discussion of conceptual challenges, evaluation details, and error analysis. The response also explains and justifies the current design choices regarding uncertainty analysis, weighting, data transparency, reproducibility, sample sizes, and the handling of segmentation and detection tasks, among other points. Overall, the response is clear and convincing, and the included summary of updates is helpful.

**Reviewer Scores:**

Taking all factors into consideration, the Area Chair believes that the scores of the two reviewers who were initially positive (both with scores of 6) would likely be maintained or improved with a full discussion period. Moreover, the two reviewers who initially expressed reservations (scores of 4 and 2) would likely raise their scores following a full discussion.

---

### Decision · Program_Chairs · 2026-01-26

Accept (Poster)